# Widespread and complex drought effects on vegetation physiology inferred from space

**Wantong Li** [1] ✉, **Javier Pacheco-Labrador** [1], **Mirco Migliavacca** [2], **Diego Miralles** [3], **Anne Hoek van Dijke** [1], **Markus Reichstein** [1,4], **Matthias Forkel** [5], **Weijie Zhang**[1], **Christian Frankenberg** [6,7], **Annu Panwar**[1], **Qian Zhang** [8], **Ulrich Weber** [1], **Pierre Gentine** [9] & **Rene Orth** [1]

The response of vegetation physiology to drought at large spatial scales is poorly understood due to a lack of direct observations. Here, we study vegetation drought responses related to photosynthesis, evaporation, and vegetation water content using remotely sensed data, and we isolate physiological responses using a machine learning technique. We find that vegetation functional decreases are largely driven by the downregulation of vegetation physiology such as stomatal conductance and light use efficiency, with the strongest downregulation in water-limited regions. Vegetation physiological decreases in wet regions also result in a discrepancy between functional and structural changes under severe drought. We find similar patterns of physiological drought response using simulations from a soil–plant–atmosphere continuum model coupled with a radiative transfer model. Observation-derived vegetation physiological responses to drought across space are mainly controlled by aridity and additionally modulated by abnormal hydro-meteorological conditions and vegetation types. Hence, isolating and quantifying vegetation physiological responses to drought enables a better understanding of ecosystem biogeochemical and biophysical feedback in modulating climate change.

Soil moisture drought is increasing in terms of duration and intensity in many areas worldwide[1]. Drought affects vegetation functioning by increasing the risk of carbon starvation and hydraulic failure which consequently induce plant mortality[2]. Since terrestrial vegetation directly regulates carbon and water fluxes at the Earth's surface, plant drought responses feedback to climate and likely aggravate global warming[3]. Vegetation function is affected by its structure (e.g., leaf demography and leaf area[4]) and its physiology (e.g., stomatal closure[5]). These two components may respond differently to environmental stress such that a comprehensive characterization of the large-scale vegetation response to droughts requires disentangling the associated structural and physiological changes.

Vegetation foliar cover emerges as one of the main properties of vegetation structure. Satellite-based vegetation greenness indices or leaf area index (LAI) products estimating green foliar cover have been widely studied to understand vegetation response to drought[6–8]. On the other hand, vegetation physiology, such as maximum carboxylation rate and stomatal conductance, have so far only been directly

[1]Department of Biogeochemical Integration, Max Planck Institute for Biogeochemistry, Jena, Germany. [2]European Commission, Joint Research Centre (JRC), Ispra, Italy. [3]Hydro-Climate Extremes Lab (H-CEL), Faculty of Bioscience Engineering, Ghent University, Ghent, Belgium. [4]Integrative Center for Biodiversity Research (iDIV), Leipzig, Germany. [5]Institute of Photogrammetry and Remote Sensing, Technische Universität Dresden, Dresden, Germany. [6]Division of Geological and Planetary Sciences, California Institute of Technology, Pasadena, CA 91125, USA. [7]Jet Propulsion Laboratory, California Institute of Technology, Pasadena, CA 91109, USA. [8]School of Geomatics Science and Technology, Nanjing Tech University, Nanjing, China. [9]Department of Earth and Environmental Engineering, Columbia University, New York, NY 10027, USA. ✉e-mail: wantong@bgc-jena.mpg.de

assessed at the site level[9]. At the same time, their changes are only implicitly included in global observations[10]. Vegetation physiology typically responds faster than the vegetation structure to environmental stressors at the ecosystem scale, yet the drought response is commonly diagnosed by concurrent structural changes which could lead to an underestimation of the vegetation functional responses[8,11–13]. In fact, drought stress first leads to reductions in stomatal conductance and maximum photosynthetic rate, which in turn reduces transpiration and photosynthesis[5,14–16]. Thereafter, vegetation structure will be reduced as a consequence of the initial physiological stress.

Recent advances in satellite remote sensing[10,17–22] bring new opportunities to monitor vegetation physiology and resulting functioning as illustrated in Fig. 1. Specifically, (i) Solar-induced chlorophyll fluorescence (SIF) is an indicator of ecosystem photosynthesis, and the TROPOspheric Monitoring Instrument (TROPOMI) onboard the Sentinel-5p satellite provides global SIF imagery continuously since 2018[17,18,23] and overcomes cloud-induced biases in previous SIF products and vegetation greenness indices[8]. As SIF is affected by photosynthetically active radiation and sun-view angular variability, we use relative SIF (SIF divided by near-infrared reflected radiance[24], hereafter 'SIFrel') as this product filters for these effects. (ii) Land surface temperature (LST)[25,26] is tightly linked with ecosystem evapotranspiration (hereafter 'ET'). Therefore, since ET cannot be directly observed at the global scale[27], we convert LST from the Moderate Resolution Imaging Spectroradiometer (MODIS) into ET using a simplified surface energy balance model[28,29] (hereafter 'SSEB'). (iii) Vegetation water content can be estimated from microwave remote sensing to assess vegetation hydraulics. High-frequency vegetation optical depths (VOD) retrievals are sensitive to upper-canopy water content changes, such that they carries information about stomatal regulation at the diurnal time scale[30]. For instance, X-band VOD from the Advanced Microwave Scanning Radiometer 2 (AMSR2) has been used to monitor ecosystem hydraulics through the ratio of midday and midnight observations[3,31,32] (hereafter referred as 'VOD ratio'). Synthesizing these opportunities can facilitate the study of global vegetation drought response from the perspective of plant physiology and enable a comprehensive diagnosis of drought effects on ecosystems globally.

In this study, we synergistically explore SIF, ET, and the VOD ratio to assess the overall and the physiological vegetation response to drought across the globe. Our study is based on data from March 2018–October 2021 at 8-daily temporal and 0.25˚ spatial resolution (Methods Section: Observation-based data) where all data products are concurrently available. We define drought events based on the soil moisture minimum during the growing season, and to focus on severe drought we only consider grid cells where the minimum of the 1982-2021 monthly soil moisture reanalysis record falls in our study period (Methods Section: Drought detection). A drought peak per grid cell is identified from 8-daily soil moisture data matching the temporal resolution of the satellite-based data streams, so that we can study the trajectories of ecosystem physiology before, during, and after drought peaks. We introduce a random forest-based approach to isolate the physiological components in SIFrel, ET, and VOD ratio (Fig. S1; Methods Section: Disentangling vegetation physiology). We determine the structural response as the variability of SIFrel, ET, or VOD ratio explained by concurrent changes in LAI in a random forest model, while the variability explained by hydro-meteorological variables in another random forest model indicates the physiological response. For this purpose, we assume that (i) LAI captures all the structural changes which are relevant for SIFrel, ET, and VOD ratio, and (ii) the physiological response can be predicted by hydro-meteorological data. Uncertainties in LAI may lead to an overestimation of the physiological estimates. Vice versa, any physiological regulation which is reflected in LAI changes within the considered 8-daily time steps will be assigned as a structural change, leading to an underestimation of the physiology estimates. This way, we can only detect 'unique' physiological variations which do not co-vary with structural changes at the considered 8-daily time scale. In addition to the remote sensing-based analyzes, we use the Soil Canopy Observation of Photochemistry and Energy flux (SCOPE) model[33] to simulate the vegetation drought response and underlying physiological changes, and hence enable a

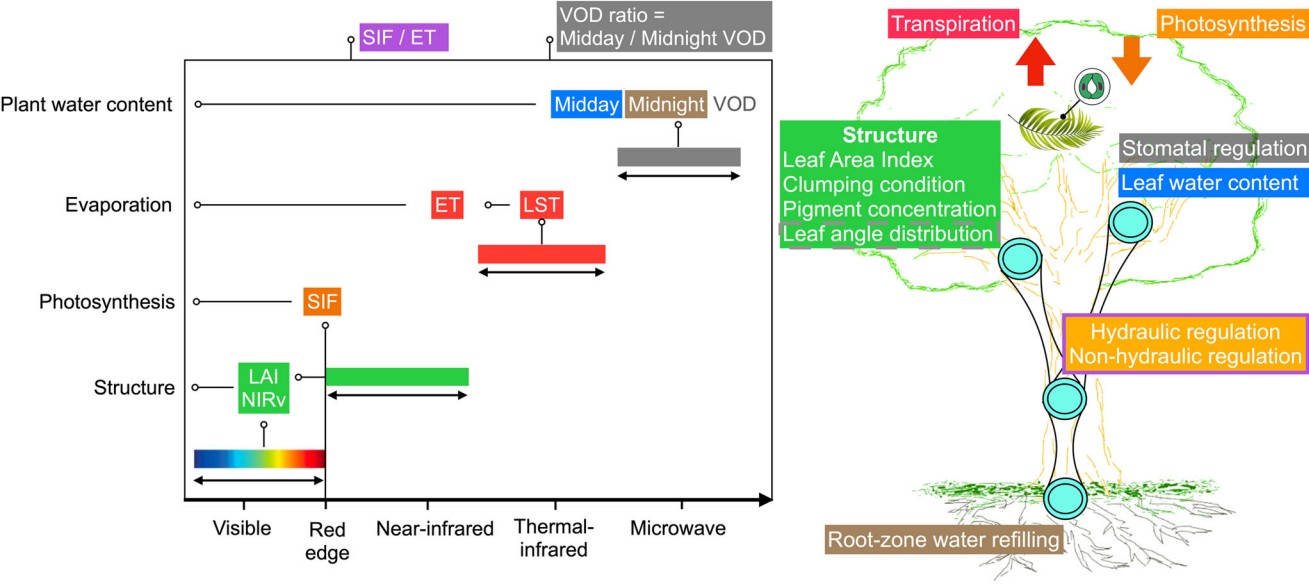

**Fig. 1 | Overview of satellite-observed wavelength bands and their information related to vegetation functioning, structure, and physiology.** Left: Considered wavelength bands and respective vegetation products employed in this study. Right: Functional, structural, and physiological aspects of vegetation dynamics, which can be inferred from the considered data. Colors of boxes indicate the link between data on the left and processes on the right. LAI: leaf area index; NIRv: the near-infrared reflectance of vegetation is an alternative product of vegetation canopy structure; SIF: sun-induced chlorophyll fluorescence; LST: land surface temperature; ET: evapotranspiration simulated from land surface temperature with a simplified surface energy balance model; VOD: vegetation optical depth. Leaf angle distribution is one of vegetation structural properties, but we note it by a dashed box because it is not globally available.

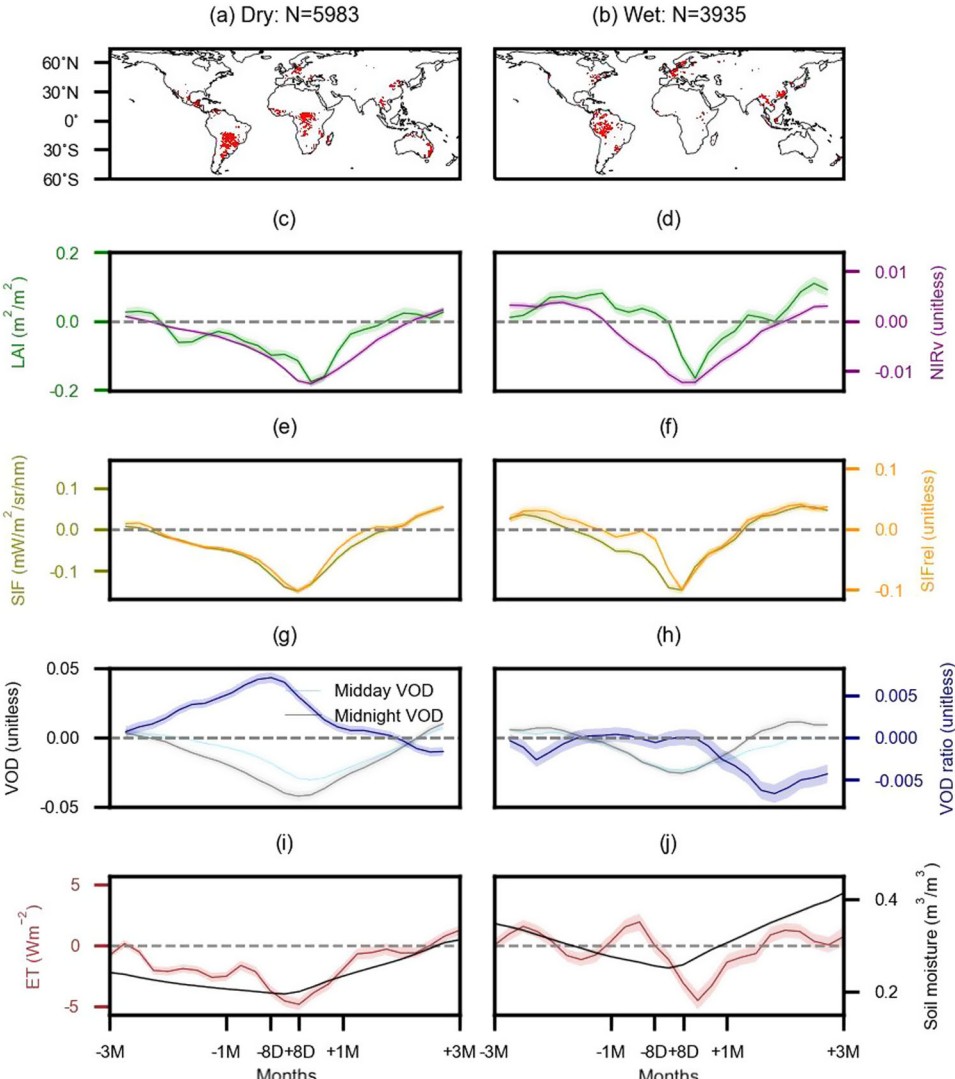

**Fig. 2 | Drought-related trajectories of multiple remote-sensing vegetation anomalies.** Drought-affected grid cells in (**a**) dry regions (aridity > 1) and (**b**) wet regions (aridity <= 1). Results for dry regions are presented in (**c, e, g, i**) and for wet regions are presented in (**d, f, h, j**). **c, d** LAI and NIRv, **e, f** SIF and relative SIF (SIFrel), **g, h** VOD at midday, midnight, and the ratio between them (VOD ratio), **i, j** ET and soil moisture. All vegetation variables are shown as anomalies, except for soil moisture in (**i, j**) which is presented in absolute values to indicate the actual water amount. Grid cells are only considered if data are available for at least 20 out of the 24 displayed time steps before, during, and after drought peaks. The solid lines denote mean values across grid cells. Shades in figures denote the mean standard error which is computed based on the standard deviation across the anomalies of every third grid cell in latitudinal and longitudinal directions, respectively. Using a subset of grid cells as opposed to all wet or dry grid cells, the effect of spatial autocorrelation is reduced.

mechanistic interpretation of our disentangled vegetation physiology (Method Sections: SCOPE simulations). By comparing model simulation results with findings from our machine learning-based analysis, we demonstrate that our methodology can isolate the vegetation physiological signals. This way, we can show that the vegetation drought response observed across data streams is largely attributable to physiological processes with the strongest downregulation in water-limited regions.

## Results and discussion
### Diagnosing the global vegetation drought response from space
Figure 2a, b shows regions around the globe where severe soil moisture droughts have occurred during our study period, distinguished between wet and dry climate regions as separated by aridity. Information on the timing of these drought events is shown in Fig. S2. Many of these events have been featured in recent literature, for instance, the 2018 European drought[34] and the 2017–2019 southeastern Australian drought[35]. Figure 2c–j presents anomalies of all employed

vegetation data streams, respectively, before, during and after the drought events, averaged across grid cells as shown in panels (a) or (b).

We present the vegetation drought response between 3 months before and 3 months after drought peaks and find a contrasting response between dry and wet regions. All vegetation and hydro-meteorological data are de-seasonalized and de-trended to minimize confounding effects (Methods Section: Data pre-processing). While LAI is below normal in dry regions due to water stress, wet regions more often show positive LAI anomalies during drought development. This is related to the drought-associated sunny conditions in these typically energy-limited regions which stimulate photosynthetic capacity, water use efficiency, and consequently vegetation greening[36,37] (Fig. 2c, d). A similar pattern is found in the case of near-infrared reflectance of the vegetation (NIRv), which is used as an alternative indicator for vegetation structure[38,39] (Fig. 2c, d). Increases in NIRv during drought development compared to non-drought years are less pronounced compared to LAI as NIRv is more sensitive to decreases in soil moisture[40,41].

Vegetation photosynthesis approximated by SIF shows continuous decreases preceding the drought peak with stronger negative anomalies in dry regions (Fig. 2e, f). As in the case of LAI, SIF increases to normal levels and even beyond after about 1–2 months after the drought peaks, and wet ecosystems recover more quickly than dry ecosystems, as the new leaf flushing of wet ecosystems promotes photosynthesis[36]. SIFrel shows similar dynamics during and after drought peaks, but in the drought development periods, SIFrel negative anomalies in wet regions are weaker. Previous studies also found that after filtering out the effects of solar irradiance and satellite viewing angles, SIFrel shows less decreases or even increases in vegetation photosynthesis during drought in Amazon forests[18,24,42].

Midday and midnight VOD anomalies both decrease during drought development and reach their lowest values shortly after soil moisture minima and then recover at a similar pace as SIFrel (Fig. 2g, h). In dry regions, midnight VOD anomalies become more negative than midday anomalies, which is not the case in wet regions. As a result, dry regions exhibit positive VOD ratio anomalies during most of the drought period, which indicates plant stomatal closure to save water as a response to high vapor pressure deficit (VPD) and soil dryness[9,14,43]. Previous studies have also illustrated similar water-saving strategies based on VOD data but mostly for seasonal time scales and long-term trends[31,44].

ET anomalies, as estimated from LST, are overall positive before drought peaks in wet regions and negative in dry regions, which is in line with previous findings using independent datasets[27,45] (Fig. 2i, j). The fluctuation of temporal ET anomalies is larger than that of other vegetation-related variables, as ET is more directly affected by hydro-meteorological variations such as changes in atmospheric water demand and soil moisture[46]. Moreover, we validate the ET inferred from LST by comparing it with ET observations from 47 eddy covariance towers. There is wide agreement between them across towers in different climate regimes and with different land cover types (Fig. S3) and also during the specific defined drought periods (Fig. S4). ET estimates during drought show slightly higher accuracy in drier than in wetter regions due to the potential strong variability in atmospheric water demand or aerodynamic conditions in the latter[46].

Contrasting vegetation responses across wet and dry regions might be related to different levels of (i) environmental stress and (ii) drought vulnerability related to different vegetation types which are investigated exclusively in the next section, and (iii) water accessibility from deeper soil layers. The absolute moisture content in the top meter of soil is higher as well as the absolute VPD is lower in wet regions than that in dry regions (Fig. 2i, j; Fig. S5i, j), making vegetation in dry areas harder to access soil water and groundwater resources but easier to transport water to the atmosphere[5,47,48]. The strongest reductions in vegetation-related variables are commonly found one time step after the actual soil moisture minimum. This is related to the time which is needed for the water from the first precipitation event after peak drought to infiltrate into the soil and to be available for plants[49]. We further quantify the spatial variability in the vegetation drought response across grid cells with the envelopes in Fig. S6 and find that this is large, underlining the relevance of vegetation and soil characteristics for the local vegetation drought response. Note that this spatial variability does not necessarily reflect the uncertainties related to the assessment of vegetation drought responses. The normalized anomalies of vegetation drought trajectories are presented in Fig. S7. The result shows a larger magnitude of NIRv and SIF anomalies compared to other vegetation variables, and soil moisture reductions show larger variability in wet than dry regions.

## Vegetation physiological response to drought

Moving beyond the full vegetation drought responses displayed in Fig. 2, we compare overall anomalies of SIFrel, ET, and VOD ratio, and their respective physiological components across aridity classes in Fig. 3. Note that seasonal cycles are computed per grid cell based on only four years of data and therefore potentially affected by individual extreme years. However, we aggregate our results of vegetation anomalies in space across many grid cells of e.g. similar aridity to improve the robustness of the results. To filter out areas with low data quality or notable human influence, we only consider regions where SIFrel, ET, and VOD ratio can be reproduced by the full random forest model considering LAI and hydro-meteorological predictors (out-of-bag $R^2 > 0$; Fig. S8). Physiology signals are derived by removing the anomalies related to structural changes, as determined from the random forest model based on LAI only (Fig. S1; Methods Section: Disentangling vegetation physiology). We find that overall and physiological patterns of vegetation anomalies across drought phases and aridity classes are largely similar, and physiological changes explain 60-97% of the overall functional drought responses in Fig. 3d, e. The physiological downregulation is generally strongest around drought peaks in sub-humid and semi-arid areas. Physiological anomalies emerge a month before the drought peak in dry regions which is earlier than anomalies emerged in other regions. Physiological changes lead to severe decreases in SIFrel and ET, whereas the VOD ratio is clearly enhanced. Increased VOD ratio suggests that drought stress leads to reduced stomatal conductance and relatively higher vegetation water content during the day than the night. The magnitudes of physiological changes in SIFrel, ET, and VOD ratio are larger than the respective structural changes (Fig. S9). In wet regions, structural and physiological changes of SIFrel have different signs which indicates the decoupling between structure and photosynthetic rate, while for the case of ET, structural and physiological anomaly patterns are similar with negative anomalies in dry regions and positive anomalies in wet regions. Structural anomalies for VOD ratio do not have a clear pattern, and the anomaly magnitude is very small, due to very few structural signals remaining in the ratio. The physiological and overall changes in VOD ratio, in turn, are very similar in terms of magnitude and directions, indicating that the original VOD ratio largely captures physiological changes. Note that the random forest model performance is rather limited when predicting anomalies of global vegetation indices compared to the prediction of time series that include the seasonal cycles[40,50]. We also present vegetation physiology patterns using higher thresholds of out-of-bag $R^2$ (i.e. 0.1 and 0.2) to further constrain model uncertainties in separating physiological signals. Results indicate the physiological anomalies are a bit more pronounced but overall largely unchanged, except for SIFrel physiology in very wet regions with an out-of-bag $R^2$ threshold of 0.2 due to the low number of available grid cells (Fig. S10).

Derived physiological changes in SIFrel, ET, and VOD ratio under drought carry different information of vegetation physiology. Physiological changes in SIFrel can largely reflect changes in the efficiency of vegetation photosynthesis which show strong decreases in sub-humid and semi-arid regions (Fig. 3d). In the case of ET, we assume that physiological changes in ET largely reflect the plant stomatal regulation (Fig. 3e). As we focus on drought periods, ET from soils and intercepted water should only have minor contributions[51,52]. Stronger decreases of stomatal conductance in dry regions suggested by ET are also confirmed by concurrent positive changes in the VOD ratio (Fig. 3f). Note that ET is not exclusively influenced by vegetation leaf area and stomatal regulation but also by the direct meteorological variability which determines the atmospheric water demand[53]. Hence, our estimate of the physiological component of ET cannot eliminate the effect of direct meteorological influence in addition to changes in stomatal regulation in vegetation physiology. Nevertheless, both VOD ratio and ET indicate stronger downregulation of physiological controls in dry regions, together suggesting the robustness of our results despite a direct impact of meteorology on ET.

By comparing midday and midnight VOD, we assume that plants are able to extract water from the soil during the night to compensate

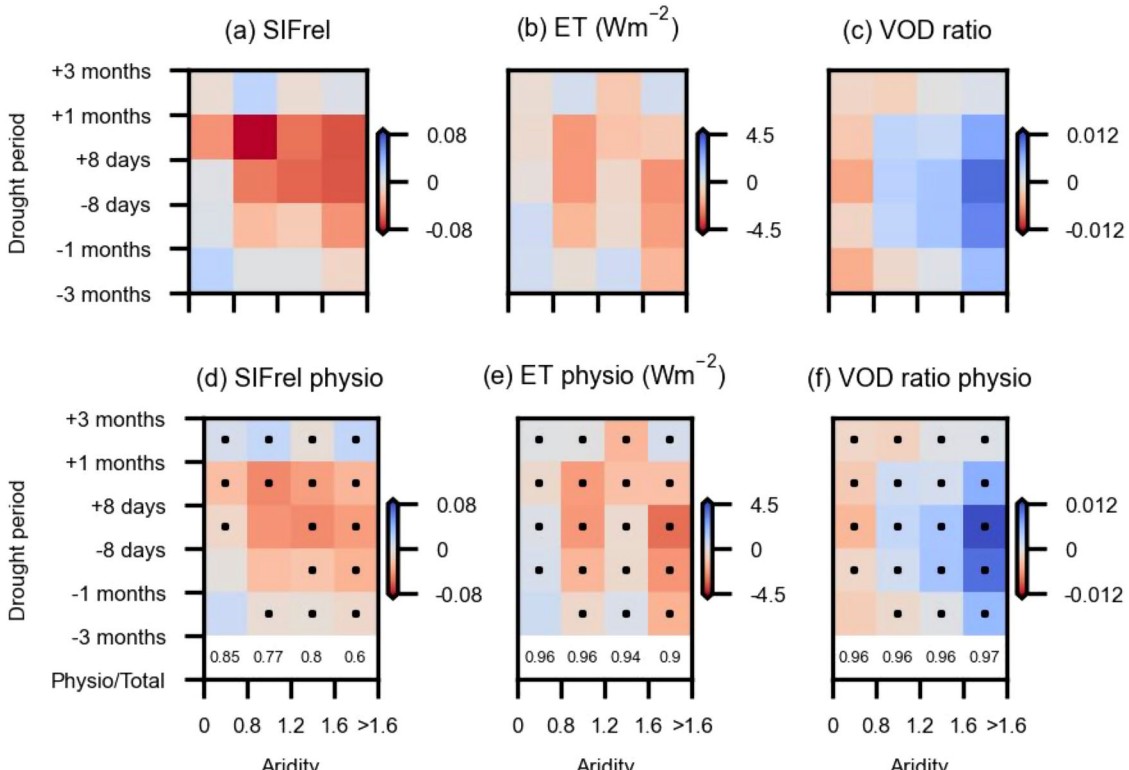

**Fig. 3 | Vegetation functional and physiological responses to drought.** Ecosystem functioning during drought as reflected by **a** SIFrel (unitless), **b** ET (Wm$^{-2}$), and **c** VOD ratio (unitless) anomalies. Ecosystem physiology (physio) is estimated as the components of **d** SIFrel, **e** ET, and **f** VOD ratio anomalies remaining after removing the LAI-related variations. Each aridity-drought period box shows the median value across corresponding grid cells and time windows. Aridity classes are chosen to yield a similar number of grid cells in each group on the x-axis. The numbers in the bottom rows denote the median ratio between vegetation physiological anomalies and total functional anomalies across the entire drought period (Physio/Total). Black dots in each bin denote that in more than 60% of the grid cells, the vegetation physiological anomaly is significantly different (95 % confidence) from a random sample of 1000 members from the same season (i.e. ± 16-day time steps) of a non-drought year.

for daytime losses, such that the nighttime plant water content would be higher and in equilibrium with the available soil moisture[31,32]. However, this is not true everywhere as we find some grid cells in semi-arid and boreal regions where absolute midnight VOD is lower than midday VOD. This could be related to sparse vegetation inducing horizontal temperature differences in these regions which are not considered in the satellite retrieval[31]. Note that these grid cells do not affect our conclusions, as similar results are found when excluding them from the analysis (Fig. S11).

While through the use of SIFrel and our approach to remove the variations related to LAI, we account for temporal changes in satellite viewing geometry, irradiance, and vegetation structure, we do not explicitly account for potential changes in the photon escape probability[54]. These could occur in response to changes in leaf clumping[55] or leaf angles. Whereas MODIS LAI includes a clumping correction[56], the leaf angle distribution is not considered, and leaf angle distribution data is not available at a global scale[57]. Moreover, uncertainties in LAI retrievals also affect our physiological estimates. To further test if using LAI could underestimate vegetation structural changes, we replace LAI by NIRv which is an alternative indicator of vegetation structure and thus can avoid the simplification of leaf angle distribution in the application of LAI[39] (Fig. S12). Overall, this yields similar patterns of physiological controls, suggesting the capacity of using LAI in representing most synchronized vegetation structural changes. Only the magnitude of our estimates is reduced throughout. This could be related to the fact that NIRv is sensitive to surface soil moisture[40,41] which is then also removed from the physiological partitioning of SIFrel, ET, and VOD ratio signals.

Next, we aim to understand the drivers of the detected physiological effects. While so far we focus on aridity, we additionally investigate the relevance of drought duration, vegetation characteristics, and selected hydro-meteorological variables. Specifically, we test to which extent anomalies of these potential drivers agree with the determined spatial patterns of physiological components of SIFrel, ET and VOD ratio. We average physiological components of SIFrel, ET, and VOD ratio across 3-month drought development and drought recovery periods, respectively. We then perform an attribution analysis using an explainable machine learning method (SHapley Additive exPlanations, hereafter 'SHAP') to quantify the relative importance of the considered variables on each of the physiological variables and drought periods (Methods Section: Attribution analysis). Overall, considered drivers in this attribution analysis can explain over 35% (cross-validation R$^2$) of the spatial variability of physiological changes from each data stream. Results show that, in general, aridity and tree cover fraction are the most relevant controls of spatial variations of vegetation physiological changes during drought development (Fig. 4a–c), even though tree cover fraction is less important in the case of SIFrel. This is confirmed when displaying the physiological anomalies of SIFrel, ET, and VOD ratio detected in the 8 days before drought peaks in aridity-tree cover fraction panels with stronger physiological controls occurring in dry areas covered with grasses and shrubs[58] (Fig. S13). In addition, meteorological anomalies also influence vegetation physiology during drought development, such as incoming shortwave radiation for SIFrel, precipitation for ET, and VPD for VOD ratio, respectively, which is in line with previous research[27,58,59]. The duration of the drought development is one of the dominant controls of the physiological component of SIFrel.

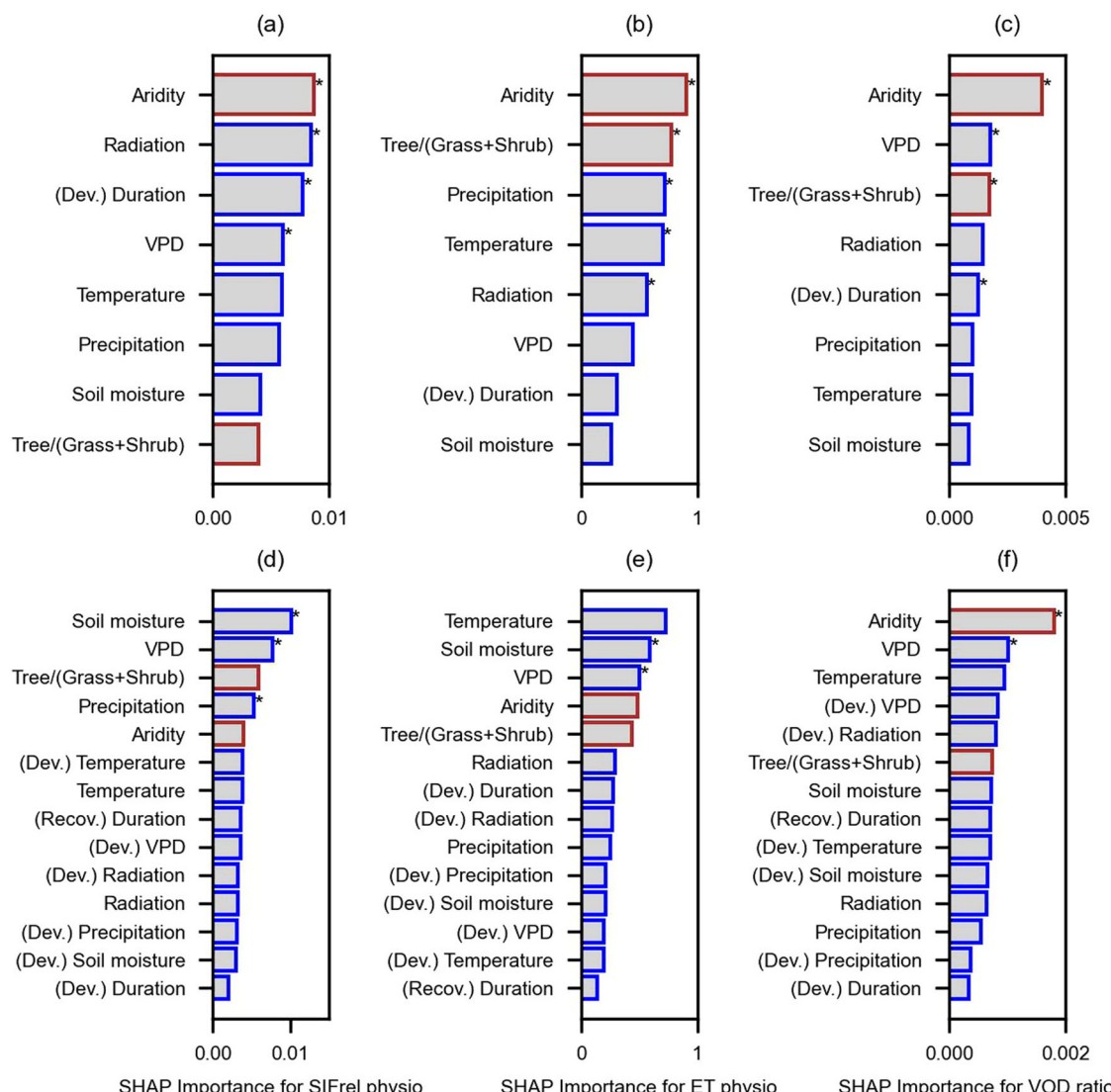

**Fig. 4 | Identifying drivers of global patterns of vegetation physiological anomalies under drought.** Considered drivers include mean climate and vegetation characteristics (in red), drought-related hydro-meteorological anomalies and drought duration (in blue). Results show their relevance in explaining the spatial variability of anomalies in **a** SIFrel physiology (SIFrel physio, unitless), **b** ET physiology (ET physio, Wm⁻²) and **c** VOD ratio (unitless) during drought development. **d–f** Similar as in (**a–c**) but for drought recovery periods where we consider drought-development (Dev.) and recovery (Recov.) related drought duration and hydro-meteorological anomalies. The unit of relative importance is the same for each physiological variable. Radiation refers to incoming shortwave radiation. * denotes the variables in the top 5 ranking in SHAP importance results are consistent with correlation results in Fig. S14.

For physiological effects during drought recovery, we consider concurrent meteorological anomalies as well as meteorological anomalies from the related drought development periods, and find that coinciding soil moisture or VPD anomalies dominantly control the spatial patterns of physiological recovery[60]. We also find that hydro-meteorological variables are relatively more important in regulating ecosystem physiology during drought recovery compared to climate and vegetation characteristics. Only in the case of VOD ratio, the physiological effects during drought recovery are mainly controlled by aridity (Fig. 4d–f). We also apply Spearman's correlation as an alternative method of assessing and ranking the variable importance to avoid the potential underestimate of the variable importance in random forests due to the multivariate collinearity. For this purpose, we compute the absolute correlation coefficient between each considered explained variable and the vegetation physiological variable (Fig. S14). We find that the first-order controls of vegetation physiology during drought development periods (i.e. aridity, tree cover fraction, and main meteorological anomaly controls) are consistently identified in

the correlation analysis, and in drought recovery periods, instantaneous soil moisture, VPD, and a few more meteorological drivers are robust in regulating spatial variability of vegetation physiology.

Furthermore, we test the robustness of our approach to isolate physiological changes from SIFrel, ET, and VOD ratio anomalies (Method Section: Disentangling vegetation physiology). In order to avoid overfitting, we leave out one every 24-time steps which is similar to the time periods displayed in Figs. 2 and 3 in training a random forest model to predict that 24-time-step result at each grid cell. Testing different time windows of 6 and 12-time steps results in similar diagnosed vegetation physiological responses to drought (Fig. S15), indicating that overfitting does not affect our results. In addition, instead of determining the physiological effects through the difference of two random forest models, (i) we apply the SHAP method to the full random forest model including LAI and hydro-meteorological variables, and consider the SHAP contribution of hydro-meteorological variables to the variations of SIFrel, ET, and VOD ratio as the physiological component. An alternative method is to (ii) fit

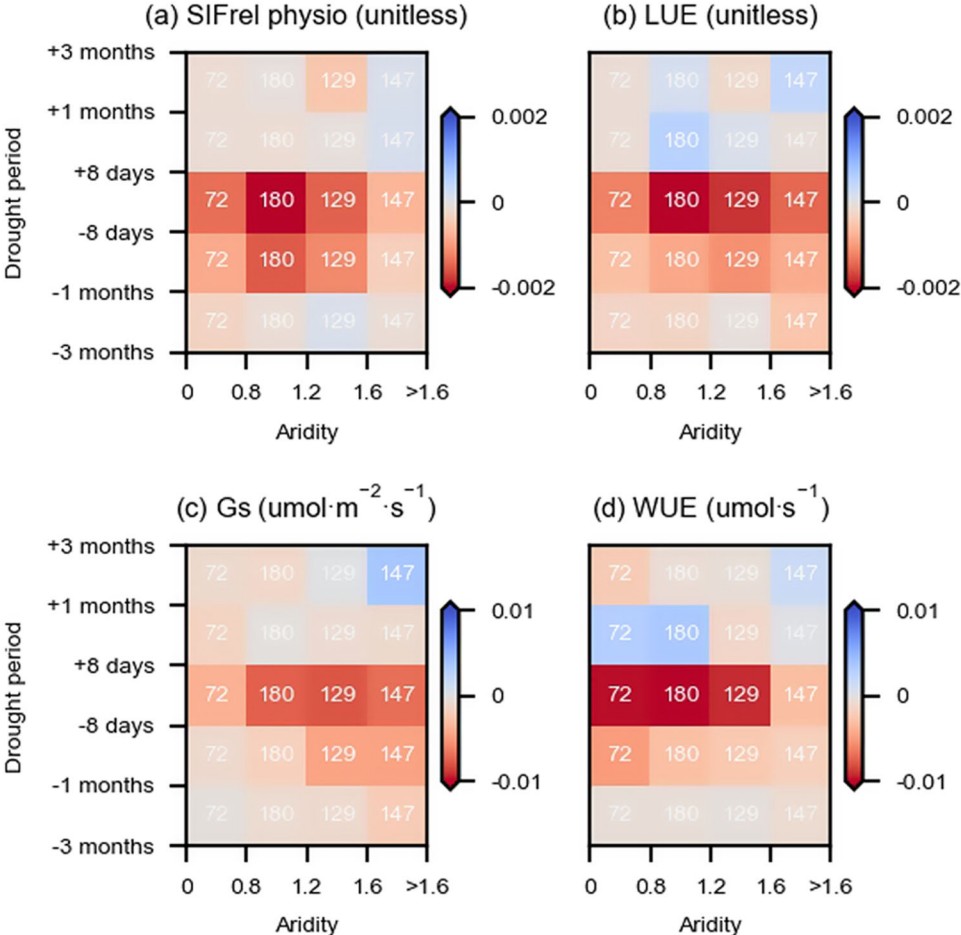

**Fig. 5 | Modeled physiological response to drought. a** Physiological component of SIFrel (SIFrel physio) as computed by the difference between dark-adapted and steady-state SIF yield; **b** Light use efficiency (LUE) as calculated by the ratio between GPP and absorbed photosynthetically active radiation; **c** Stomatal conductance (Gs); **d** Water use efficiency (WUE) as calculated by the ratio between GPP and ET.

Note that instead of the entire study domain, this analysis is based on 600 randomly chosen grid cells considered in the observation-based analyzes and distributed across the globe. Observation-based results for the same grid cells are presented in Fig. S20. The white numbers denote numbers of grid cells belonging to a certain aridity group.

a multivariate regression model instead of random forests and isolate explained variations related to the hydro-meteorological variables as physiology. Both approaches (i) and (ii) yield similar results (Fig. S16), and support findings of strong physiological regulations in sub-humid and semi-arid regions using the main method (Fig. 3).

The vegetation drought response may be affected by preceding drought events. To test if this affects our results, we repeat our global analysis without grid cells where the second-strongest drought occurs in our study time period in addition to the strongest drought. The results with the reduced set of global grid cells are shown in Fig. S17. We find patterns similar to Fig. 3, indicating that multiple drought events have no major impact on our analysis. To further test the robustness of the drought detection, we (i) more strictly select severe drought events by checking if the detected driest soil moisture is lower than a threshold of −1.5 standard deviations below the seasonal mean value of the entire 40-year soil moisture and (ii) detect drought peaks using minimum soil moisture anomalies rather than absolute soil moisture. Figure S18 shows that with a more strict severe drought evaluation method, the remaining grid cells can largely reproduce physiological patterns of SIFrel, ET, VOD ratio under drought. In the case of using soil moisture anomalies to detect drought in Fig. S19a, b, the detected drought years are largely the same while the seasonal timing differs a bit in some regions. The patterns of estimated physiological drought responses (Fig. S19c–e) are similar even though with smaller

magnitudes, as the absolute soil moisture can be higher and vegetation water stress can be reduced in the case of detecting drought by the driest soil moisture anomalies when comparing it to the case of detecting drought by the driest absolute soil moisture.

**Mechanisms underlying the physiological response to drought**
In addition to the observation-based analysis, we employ SCOPE, a soil–plant–atmosphere continuum model coupled with a radiative transfer model, to perform and study simulations of the vegetation response to the identified drought. This allows us to mechanistically understand the diagnosed physiological signals from observations. We choose the SCOPE model[33] as it considers vegetation physiology and produces radiance spectra associated with vegetation functioning and biophysical properties (Method Section: SCOPE simulations). Similar to observation-based analyzes, SCOPE uses LAI and hydro-meteorological data as inputs. Note that given the considerable computational effort, we randomly selected a subset of 600 grid cells distributed across the globe for performing SCOPE simulations and analyzes. Observation-based physiological effects are similar in this subset of grid cells to those from all previously considered areas in terms of the overall contrasting patterns of vegetation physiology between wet and dry regions during the drought development period (Fig. S20; Fig. 5d–f), while sub-arid and arid regions show slight differences when comparing model results with observations due to potential uncertainties of model structure and limited sampling data.

The physiological SIF response to drought from SCOPE is calculated by the difference of the SIF yield between dark-adapted and light-adapted plants (Fig. 5a). Dark-adapted SIF represents an unstressed reference used by the model which is independent of the environmental conditions. The light-adapted SIF responds to environmental stresses such as light saturation and high VPD[4,33]. The inferred physiology component of SIFrel is similar to the observed SIFrel physiology pattern during drought development and drought peak periods with strong decreases in sub-humid and semi-arid regions (aridity between 0.8–1.6). It is related to the modeled light use efficiency (LUE) anomalies which show the same pattern. However, physiological reductions recover more quickly in the model than in observational estimates after drought peaks. This can be explained as SCOPE does not account for drought stress through soil moisture deficits and respective legacy effects, thus the vegetation in our simulations is mechanistically only affected by atmospheric dryness[33]. While this is a disadvantage for simulating the drought recovery, it is justified for the simulation of drought development and drought peak periods as (i) high VPD is typically strongly related to soil moisture drought stress[46], and (ii) because our observation-based analysis shows that VPD plays a more important role than soil moisture in regulating physiology during drought development (Fig. 4a–c). The simulated stomatal conductance shows decreases in transitional and dry regions, which agrees with observed physiological changes in ET and VOD ratio anomalies. The strong stomatal regulation in dry regions is consistent with comparatively lower decreases in water use efficiency (WUE). Our derived contrasting patterns of WUE between wet and dry regions are in line with previous research but not the signs of WUE changes[60]. Overall, the SCOPE results provide clear evidence of vegetation physiological responses to drought in support of our observation-based findings. Additionally, the simulations allow us to mechanistically attribute the observed physiological signals to underlying changes in stomatal conductance, water use efficiency, and light use efficiency.

In this study, we take advantage of recent advances in satellite remote sensing and machine learning applications to improve the understanding of vegetation responses to drought. In particular, we isolate the physiology-driven effects of drought on vegetation functioning at the global scale. We find that the observed drought responses of photosynthesis and evapotranspiration are related to, and largely driven by, unique changes in vegetation physiology. The vegetation's physiological response to drought is most pronounced in transitional and semi-arid areas, and for shrubs and grasses. Despite the fact that our method simplifies vegetation structural changes and cannot separate potentially direct meteorological-driven signals beyond physiology in the case of ET, the robustness of our result is addressed by synthesizing multiple observations. The physical-based model SCOPE suggests similar patterns in the diagnosed physiological changes during drought. The model additionally allows us to understand the underlying processes that the downregulation of stomatal conductance and light use efficiency are highly relevant to determine the vegetation response to water stress. Overall, we have been able to quantitatively disentangle vegetation physiology by synthesizing multiple state-of-the-art remote sensing data streams. This is essential given the present uncertainties in simulated large-scale ecosystem drought responses[11]. In this context, disentangling physiological and biophysical vegetation responses enables better characterization of these distinct vegetation response pathways to consequently reflect their interplay more accurately in Earth system models through better parameterizations of vegetation physiology such as photosynthetic capacity[10], or through upgrading the respective model structure such as incorporating sufficient soil water stress on vegetation physiology[61]. This will support the accuracy of their simulated land-climate feedbacks in particular, and of their climate projections in general.

## Methods

### Observation-based data

We combine satellite remote-sensed Leaf Area Index (LAI), the near-infrared reflectance of vegetation (NIRv), solar-induced chlorophyll fluorescence (SIF), land surface temperature (LST), and Vegetation Optical Depth (VOD) data to investigate global vegetation drought responses (Fig. 1). Among them, we use ungridded TROPOMI SIF resampled to the 0.25° spatial resolution and available from March 2018 to October 2021. SIF is then normalized by near-infrared reflected radiance (relative SIF) to account for bidirectional reflectance effects and incoming solar irradiance[24]. Although previous studies indicate rare cloud cover influence on the SIF seasonality, cloud influence cannot be completely ruled out[18]. Therefore, TROPOMI data are pre-filtered to remove soundings covered by heavy clouds (over 0.5 of cloud fraction) to lessen potential cloud effects and maximize the number of observations during wet seasons.

Moderate Resolution Imaging Spectroradiometer (MODIS) products MOD15A2H LAI (8-daily), and MCD43C4 NIRv (daily; a product of NDVI and the near-infrared reflectance) are used as proxies of vegetation structure or abundance. MODIS Land Surface Temperature (LST) from the MYD11C1 product is used to estimate daily ET. Good-quality data from MODIS products are ensured by quality flag filtering. LST daily data are used in combination with a simplified surface energy balance model (SSEB) to estimate ET at the daily scale[28,29]. Due to not available global ET measurements, several global observation-based model-driven ET products have been developed[62]. However, we do not consider these ET products because they commonly show less accuracy in extreme events where physiological changes prevail. Therefore, we use daily MODIS LST data (with overpassing time at 13:30 p.m.) and ERA5-Land meteorological variables combined with the SSEB model to estimate ET. Additionally, hourly SSEB model meteorological inputs (including air temperature, incoming shortwave and longwave radiation, surface pressure, atmospheric vapor pressure, soil moisture, and wind speed) are obtained from ERA5-Land reanalysis[63]. Note that we adapt SSEB from Panwar & Kleidon (2022)[28] by the following two simplifications: (i) Daily minimum surface and air temperatures are assumed to be equal; (ii) Daily maximum LST is represented by MODIS LST at 13:30 p.m. Therefore, daily ET is calculated by:

$$ET = \left(1 - \frac{C_p \cdot \rho \cdot (LST_{max} - Ta_{max}) \cdot (1.4 g_a)}{1.4 Rs_{max}}\right) \cdot Rn_{mean} \quad (1)$$

where ET is estimated by latent heat flux (Wm$^{-2}$) (hereafter 'ET'), $LST_{max}$ is MODIS land surface temperature at 13:00 p.m. (K), $Ta_{max}$ is maximum hourly air temperature (K), $Rs_{max}$ is maximum hourly net shortwave radiation (Wm$^{-2}$), and $Rn_{mean}$ is daily mean net radiation (Wm$^{-2}$). $C_p$ (1005 J kg$^{-1}$ K$^{-1}$) is specific heat capacity and $\rho$ (1.23 kg m$^{-3}$) is the simplified density of air. $ga_{mean}$ (m s$^{-1}$) is mean diurnal variation of aerodynamic conductance. As summarized in Panwar & Kleidon (2022), $g_a$ is different across vegetation types with higher values in forest ecosystems and lower values in grass ecosystems. We distinguish tree, short vegetation (grass and shrub), and soil, and depending on their respective fractional cover, $g_a$ is set to 0.06, 0.0345, and 0.002 m s$^{-1}$, respectively. Among them, the vegetation-related values are from Panwar & Kleidon (2022)[28] and the soil-related value is from van der Tol et al. 2014[33].

We evaluate the ET at a 0.25° spatial resolution using 8-daily eddy-covariance-measured latent heat for 73 sites, which are from the ICOS-drought-2018 dataset. The raw data are processed following the ONEFLUX pipeline[64]. Latent heat flux is first gap-filled using marginal distribution sampling[65] and then aggregated to the 8-daily scale by computing the mean value. We exclude the days when more than 20% value is missing for each 8-day window to eliminate the potential resampling-induced noise. Gridded ET data are matched to each site using the nearest neighbor method (Python version 3.9.7; xarray

version 0.20.1). In total, 47 sites are selected since they offer sufficient data. We show the distribution of the correlation coefficient between raw data of EC-measured LE and ET across each site and find a median correlation of 0.88 for the whole growing season (Fig. S3) and 0.8 for drought-specific periods during 3 months before until after drought peaks (Fig. S4).

Microwave remote sensing provides great opportunities to monitor vegetation water content and has increased applications in drought-related ecological studies[3,30,66]. Global multi-frequency VOD products are currently available. They are irrespective of cloud cover, and high frequencies are more sensitive to the upper part of canopy changes. Since we aim at studying vegetation physiological dynamics, X-band VOD at 10.7 GHz from LPDR v2 is used, as it is sensitive to upper-canopy water content, which relates to the stomatal regulation[67]. LPDR X-band VOD from AMSR2 sensors for the period of 2018-2021 has day-night observational capabilities, although its spatial resolution at 0.25° is relatively coarser than VOD data from sun-synchronous orbits and than other vegetation observations such as TROPOMI SIF and MODIS bands (for which 0.05° is available to use). Both the daytime and nighttime VOD contain information about vegetation water content, which scales with above-ground biomass and relative moisture content[30]. Vegetation water content can be used to represent leaf water potential and associated plant hydraulics[32]. Daytime VOD (overpassing time at 13:30 p.m.) is normally regulated by plant hydraulics caused by the imbalance between transpiration and root-zone water supply. Nighttime VOD (overpassing time at 1:30 a.m.) is driven by root-zone soil moisture refilling and is almost linearly linked to soil water potential[32]. Given pre-dawn equilibrium between leaf water potential and root-zone water potential, a combination of midday and midnight VOD can largely reduce structural variations and can be used to investigate the ecosystem isohydricity[31,44]. Here we calculate the ratio between midday and midnight VOD (VOD ratio), and investigate the responses of VOD ratio to soil moisture to characterize ecosystem isohydricity changes under drought[32]. We acknowledge that regions with incomplete root-zone water refilling at 1:30 a.m. do not show a near-linear relationship between nighttime VOD and soil water potential, which leads to spurious biases on VOD ratio. We minimize this effect by excluding regions that show greater midday than midnight growing-season averages of VOD[31], as shown in Fig. S11.

Concomitant hydro-meteorological data from ERA5-Land reanalysis include air temperature, incoming shortwave radiation, vapor pressure deficit, 1-m soil moisture, and precipitation[63]. A total of one meter soil moisture is calculated based on three layers of ERA5-Land soil moisture weighted by layer thickness of 7, 21, and 72 cm. Climate regimes are defined by the aridity index, which is calculated as the ratio of the 2018–2021 mean net radiation and unit-converted precipitation from ERA5-Land, with higher values meaning drier climate regimes.

## Data pre-processing
Fig. S1 presents a flowchart of data pre-processing and further analysis. All vegetation and hydro-meteorological data from March 2018 to October 2021 are aggregated into the 0.25° spatial resolution and 8-daily temporal resolution where all data are available. To minimize the noise in daily SIF, ET, and VOD, and to match 8-daily LAI and NIRv we produce 8-daily data from a 16-day average moving window (TROPOMI's revisiting cycle) with 8-day overlap. Windows with gaps larger than 20% are set as no data in aggregated results. TROPOMI features different overpassing daytime within the 16-day cycle, and averaging this helps to lessen the effects of sun-view geometry variability[18]. Moving window averages are also applied for all the other vegetation and hydro-meteorological reanalysis data to keep the consistency. All data are used in the form of anomalies, as we are exclusively interested in abnormal behavior of vegetation drought responses than the seasonality. For this, mean seasonal cycles are calculated for each month (January to December), and trends are

derived by using a locally-weighted smoothing function with 40% overlapping moving windows from March 2018 to October 2021, which are then removed from the time series to extract the anomalies.

This study focuses on the regions characterized mostly by significant vegetation cover and without dense human activities. For this, we remove regions having sparse vegetation cover (<5%) and a high irrigation fraction (>10%). The vegetation cover is calculated as the sum of trees, shrubs, and grasses cover fractions from ESA CCI Land-cover v2.1.1 dataset from the year 2020. The irrigation fraction was collected around the year 2005[68]. Vegetation fractional cover data is also used to distinguish tree dominant, and shrub-plus-grass-dominant ecosystems by the ratio between tree/(shrub and grass) with a threshold of 0.5.

## Drought detection
We study vegetation responses to drought during growing seasons, which are defined with temperatures higher than 5 °C and mean seasonal cycles of SIF higher than 0.2 mW m$^{-2}$ sr$^{-1}$ nm$^{-1}$. The reason to use mean seasonal cycles of SIF to account for growing seasons instead of dynamic SIF values is to largely keep vegetation post-drought anomalies, e.g., in savannas. For growing-season data, we evaluate soil moisture dryness across grid cells during 2018–2021 using 40-year ERA5-Land soil moisture reanalysis. For this purpose, we calculate yearly minima using monthly soil moisture records and rank these yearly minima, since long-term soil moisture with coarse temporal resolutions of monthly compared to 8-daily is more representative of drought severity. We focus on the grid cells where each yearly minimum occurs in 2018–2021, so that severe drought events could exist in these regions during this recent time period. Drought peaks are detected based on the lowest 8-daily soil moisture during 2018–2021 for each grid cell. We then study the vegetation data anomalies during the course of drought from 3 months before until 3 months after drought peaks. The drought duration is used to attribute the spatial variation of vegetation anomalies, and is defined by the number of time steps of soil moisture anomalies back to zero or positive values before and after drought peaks, respectively.

## Disentangling vegetation physiology
Since vegetation physiology responds significantly to drought stress, we adapt an existing approach to disentangle physiological influence from LAI-driven structural changes[69,70]. We use the way of disentangling the physiological component of SIFrel as an example. Due to an existing nonlinear relationship between SIFrel and LAI, we fit a random forest regression model to account for non-linearity[71] to predict SIFrel using LAI as the only predictor per grid cell. The predicted drought-period SIFrel is hence expected to present the SIFrel structural component. A leave-out strategy of model training is applied to avoid potential over-fitting due to the relatively straightforward power of random forest modeling but limited input information. In this way, the drought-period data are excluded from the training model when predicting drought-period SIFrel, while the model can still learn SIFrel–LAI relationships under dry conditions from the non-drought years and extrapolate such relationships to drought periods. However, note that the random forest model might be not able to predict extreme values accurately, resulting in a less significant variation of vegetation structural changes. The leave-out window in the main result is defined as 24-time steps (192 days), while results using 12 and 6-time steps are shown in Fig. S15 with similar patterns of significant negative physiological changes in dry regions.

To account for potential observational noise in predicted variables (e.g., SIFrel) which are supposed to have lower signal-to-noise ratios than greenness indices and reanalysis data, a second random forest model is built to fully consider inputs including LAI and hydro-meteorological anomalies using the whole growing-season data. This model is used to predict SIFrel anomalies during drought periods, to

provide more reliable SIF variations while lessening data noise. The derived out-of-bag (OOB) $R^2$ scores from cross-validation are used to evaluate the model performance, with lower values indicating larger difficulties in predicting SIF due to bad data quality or human activities such as tree logging. Hence, regions with OOB $R^2$ lower than zero are disregarded for disentangling vegetation physiology (Fig. S8). Finally, SIF physiological components can be extracted per grid cell as the difference between the SIF anomalies predicted as a function of structure and hydro-meteorology, and as a function of the structure only.

Same steps are applied to disentangle physiological components of ET and VOD ratio using LAI and random forests. NIRv is additionally used to replace LAI as an indicator of vegetation structure in the random forest model. Using NIRv confirms our main findings that transitional-to-dry regions show strong downregulation of vegetation physiology. While NIRv better accounts for the escape probability of SIF, physiological components of SIF, ET, and VOD results all present reduced variations. This is related to the potentially confounding information of using NIRv as a structure proxy since NIRv largely synchronizes soil moisture dynamics[40,41].

Apart from trusting the random forest extrapolation ability in our analysis, we test two alternative methods using the variance decomposition. First, we use a Multiple Linear Regression model (MLR) and treat LAI and hydro-meteorological data as predictor variables of SIFrel regardless of drought or non-drought periods. The sum of variance explained by hydro-meteorological data only from the MLR for the drought period can be treated as the SIFrel physiological component under drought. In addition, we use a random forest model which accounts for nonlinear relationships instead of MLR and apply explainable machine learning[72] (SHapley Additive exPlanations, hereafter 'SHAP') to disentangle hydro-meteorological contributions on SIFrel during drought as the SIFrel physiological component. These two methods are also applied to the cases of ET and VOD ratio. Results from these two alternative analyzes support findings using our main methodology (Fig. S16). Still, we acknowledge that decomposition methods might underestimate structural components due to the collinearity between vegetation structure (i.e., LAI) and hydro-meteorological anomalies, and due to larger numbers of predictor variables to account for physiological influence compared to structural influence. We test the effect of using a lower number of hydro-meteorological predictors in the variance decomposition method and find a reduced magnitude of resulting vegetation physiological patterns for ET. This suggests that the decomposition method is sensitive to numbers of predictors (Fig. S21). Note that our main method of detecting physiological effects as the difference between two random forest models used throughout the manuscript mitigates this potential issue.

## Attribution analysis

We conduct an attribution analysis to better understand potential drivers of vegetation physiology under drought spatially. We select multiple variables related to land-surface climate and vegetation characteristics, drought-specific hydro-meteorological variables, and drought duration in explaining vegetation physiology. We train a random forest model and use these considered variables as predictors to predict SIFrel, ET, and VOD ratio-related physiology, respectively, across all study grid cells[71]. Using cross-validation out-of-bag $R^2$, we evaluate the model sufficiency in explaining physiological variations. Then, we use the SHAP values to quantify the marginal contributions of each predictor variable and identify the relative importance among different variables by ranking the averaged absolute SHAP values[72]. The attribution analysis can generally explain over 35% of the spatial variability of each physiological variable. The remaining 65% that cannot be explained by the random forest model are potentially related to uncertainties in observations of leaf area index, vegetation

photosynthesis, evaporation, and vegetation water content, and also to uncertainties in the hydrometeorological reanalysis data. Furthermore, different availability and accessibility of deep water sources such as groundwater, for which no reliable global gridded observations are available in terms of the spatial-temporal scales of our study to our knowledge, can introduce uncertainties here.

## SCOPE simulations

The state-of-the-art model the Soil Canopy Observation of Photochemistry and Energy flux (SCOPE) (v 1.73) couples radiative transfer, energy balance, and photosynthesis submodels to predict vegetation carbon, water, and energy exchanges with spectroradiometric variables directly linked to physiology (e.g., SIF, or thermal radiance). SCOPE can be used to interpret remote sensing observations[33] and test physiological assumptions[4]. SCOPE predicts photosynthesis as a function of plant traits, irradiance, leaf temperature, and other meteorological conditions using Farquhar and Collatz equations for C3 and C4 plants separately. The modular nature of SCOPE allows for separately simulating dark-adapted fluorescence and light-adapted fluorescence[33]. The difference between both radiances is that only the light-adapted fluorescence is modulated by physiology in response to environmental conditions. This fact has been successfully exploited in former studies to assess methods separating structural and physiological information from SIF time series[4].

The aim of SCOPE simulations is not to accurately reproduce observations, but to produce a comparable variability of vegetation responses to drought and remote sensing view-angle configurations that assess the validity of the metrics and analyzes applied to observations. In this context, simulations provide: (i) vegetation functioning (i.e. photosynthesis); (ii) vegetation physiology (i.e. stomatal conductance), LUE, and WUE; (iii) physiology-driven relative SIF to resembling TROPOMI data computed as the difference between light-adapted (physiologically-regulated) and dark-adapted SIF[4], and then normalized by the reflected radiance at 740 nm. We use the difference between dark-adapted and light-adapted fluorescence to validate our approach capability to disentangle SIFrel physiology rather than applying the random forest approach in the SCOPE outputs, as the separation of plant physiology in SCOPE is more mature in the former approach. Also, other vegetation physiological patterns from model outputs such as stomatal conductance and light use efficiency reasonably support our disentangled physiological regulation from vegetation observations under drought. SCOPE outputs are simulated firstly as leaf-level values and then are integrated over the canopy layer, and they are then averaged every 8 days, deseasonalized, de-trended, and used to study vegetation physiology corresponding to defined drought periods as in observation-based results.

One simulation per day is run at different times of the day, between 9 and 14 h for TROPOMI and at 13:30 for MODIS LST, allowing for a plausible variability of the view angles of each sensor. We use hourly hydro-meteorological data from ERA5-Land reanalysis as SCOPE inputs from March 2018 to October 2021. Hydro-meteorological data include 2 m air temperature, incoming short-wave radiation, incoming longwave radiation, surface pressure, atmospheric vapor pressure, 10 m wind speed, and 1 m soil moisture. 1000 grid cells are initially randomly selected based on the regions with severe drought events (Methods Section: Drought detection) for SCOPE to run. Simulations are also informed with 8-daily MODIS LAI, NDVI, and daily LST. 8-daily products are linearly interpolated to daily values for the simulations, and the simulation results are aggregated coherently with the observational datasets (See Methods Section: Data pre-processing). After removing grid cells with bad prediction performance in terms of SIFrel, ET, and VOD ratio in observation-based analysis (See Methods Section: Disentangling vegetation physiology), around 600 grid cells remain to compare the behavior of observations

and SCOPE simulations. Simulations are performed at 0.25-degree spatial resolution.

For each grid cell, we account for the spatial variability of plant functional types (PFT) by simulating them separately. While most vegetation parameters are PFT-specific, the most relevant for the simulations are adjusted to make simulations more plausible. We obtain PFT fraction cover from ESA CCI Landcover v2.1.1 dataset including broadleaf evergreen trees, broadleaf deciduous trees, needle-leaf evergreen tree, needle-leaf deciduous trees, broadleaf evergreen shrubs, broadleaf deciduous shrubs, needle leaf evergreen shrub, needle leaf deciduous shrub, natural grass, managed grass and bare soil. First, we estimate each PFT-LAI by scaling daily MODIS LAI cell input by a specific PFT average LAI and its corresponding fraction in the grid cell. The seasonal variability of the evergreen PFTs is reduced by a factor of 0.2 and the difference is distributed within the deciduous PFTs of the cell. Secondly, leaf chlorophyll content is similarly set but based on PFT-characteristic values from Croft et al. (2020)[73]. However, the grid-averaged chlorophyll content is unknown and thus estimated by assuming a saturating relationship with NDVI. We select ten timestamps of the time series evenly covering the NDVI range and numerically optimize the parameters of that function by minimizing the difference between MODIS NDVI and the predicted NDVI PFT-weighted average. At each iteration, PFT-chlorophyll content is estimated as described before and carotenoids content is set as the 35% of it. To avoid anomalous pigment values and soil reflectance, the soil bright parameter of SCOPE at each site is also optimized. For this, whenever NDVI is negative, chlorophyll content is set to zero and an additional parameter determining the fraction of snow cover is allowed to be larger than 0. Soil reflectance is then linearly mixed with snow spectra from the USGS Spectral Library. Finally, to better simulate energy partitioning in sparse vegetated areas, we constrain the relationship between soil moisture content and soil resistance when estimating evaporation from the pore space with MODIS LST observations, because soil resistance strongly controls soil evaporation and affects vegetation transpiration and photosynthesis[74]. As before, we select five points of each grid cell time series with the lowest averaged LAI that evenly covers the LST range. Then we optimize the decaying relationship between soil moisture-soil resistance by minimizing the difference between predicted and MODIS LST.

SCOPE simulation outputs are averaged according to PFT cover fraction and then aggregated every 8 days to match the processing and sun-view variability of the observational dataset. SCOPE v1.73 does not impose any photosynthesis limitation as a function of soil moisture. However, we optimize soil resistance for evaporation from the pore space, which limits soil latent heat flux in dry soils and affects vegetation. We also test applying a soil moisture constraint on maximum carboxylation rate using the empirical relationship of Bayat et al. 2019[75], but this approach has only been tested in dry ecosystems and overestimates photosynthetic stress in ecosystems with access to deep soil water layers.

## Data availability
TROPOMI SIF is available at https://web.gps.caltech.edu/~cfranken/data.html. MOD15A2H LAI is from https://lpdaac.usgs.gov/products/mod15a2hv006/. MCD43C4 NIRv is from https://lpdaac.usgs.gov/products/mcd43c4v006/. MYD11C1 land surface temperature is from https://lpdaac.usgs.gov/products/myd11c1v006/. The ICOS-drought −2018 dataset can be downloaded from https://www.icos-cp.eu/data-products/YVR0-4898. LPDR v2 X-band VOD is available at http://files.ntsg.umt.edu/data/LPDR_v2/. ERA5-Land reanalysis data are from https://cds.climate.copernicus.eu/cdsapp#!/dataset/10.24381/cds.e2161bac?tab=overview. ESA CCI Landcover v2.1.1 is from https://www.esa-landcover-cci.org/. The irrigation fraction dataset is from http://www.fao.org/aquastat/en/geospatial-information/global-maps-irrigated-areas/latest-version/.

## Code availability
The codes required for reproducing the results and figures in the main text have been deposited at https://doi.org/10.5281/zenodo.7971319, as well as the data to run the codes are available at https://doi.org/10.5281/zenodo.7971170

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

## Acknowledgements

The authors thank Axel Kleidon, Alexandra G. Konings, Benjamin Dechant, Jacob Nelson, Martin Jung, and the Hydrology-Biosphere-Climate Interactions Group at the Max Planck Institute for Biogeochemistry for fruitful discussions. W.L. acknowledges support from the International Max Planck Research School for Global Biogeochemical Cycles. R.O. and A.J.H.v.D. are funded by the German Research Foundation (Emmy Noether grant number 391059971). Q.Z. is funded by the National Natural Science Foundation of China (Grant number. 42071392).

## Author contributions

W.L., R.O., and C.F. conceived the study concept. W.L., R.O., M.M., M.R., M.F., and C.F. developed the methods and designed the experiment. W.L. conducted data analyzes, visualized results, and wrote the manuscript with suggestions from R.O., D.M., and A.J.H.v.D. J.P.J conducted the SCOPE experiment with the suggestions from M.M. W.Z. contributed to eddy covariance data processing. A.P. derived the equation of the simplified surface energy balance model. U.W. contributed to a large part of pre-processing of observation-based datasets. W.L., R.O., and J.P.J. revised the manuscript based on critical comments from M.M., D.M., A.J.H.v.D., M.R., M.F., and P.G. throughout the manuscript. D.M., A.J.H.v.D., A.P., and Q.Z. contributed to text corrections and results interpretation.

## Funding

## Competing interests

The authors declare no competing interests.
