## [Peer Review File · Nature Communications]

Widespread and complex drought effects on vegetation physiology inferred from spaceREVIEWER COMMENTS

Reviewer #1 (Remarks to the Author):

This paper investigates the vegetation physiological response to drought conditions using observational data. The manuscript is very well written, and the methodology section is detailed. I recommend the manuscript for publication with minor revisions.

This study uses a comprehensive set of remote sensing data to infer vegetation structural properties and physiological ones, and disentangle them from each other. Using soil moisture data from ERA5 they identify drought periods across the globe, and then apply their methodology to detect vegetation phenological responses. The methodology uses two Random Forest models applied to the observations and is very well illustrated in Figure S1 in addition to the description. To ensure results are not sensitive to the choice of model, authors also 1) use a Multiple Linear Regression model and 2) apply explainable machine learning technique to the random forest model, and results are consistent with the initial approach.

In addition, the SCOPE model is employed coupled with a radiative transfer model to simulate vegetation response to the identified drought conditions, and the output of the model (and the identified mechanisms) are compared to the results from observations (which validates the results). Authors then discuss their findings (Figs 2-4) in terms of vegetation physiological responses to drought, and provide physical explanation on why these results are meaningful (reference to other relevant studies that have looked similar problems are also provided and similarities and differences are discussed).

Overall, the methodology is solid, and the results are justifiable. Certainly, there are assumptions made in the data pre-processing, the methodology and what you can infer from the observations, but they are all reasonable and very well explained (including any caveats). I do not believe that these assumptions would impact the results in any way.

Minor comments:

- Lines 608-611: This is a key assumption. Please elaborate why you decided to include more predictors to account for the physiological influence (beyond the fact that the data is available)? What if you lower the number of predictors for the physiological response, how this might impact the results? If you have done any sensitivity test on this, please report.
- Proofread the paper. There are some typos here and there.
- It's encouraging to see the code will be shared for the final publication. I encourage you to share the full end-to-end code from data processing to the final results. Being able to reproduce these results is huge benefit to the community.

Reviewer #2 (Remarks to the Author):

Summary:

This is a review of the manuscript Widespread and complex drought effects on vegetation physiology inferred from space (NCOMMS-23-08597) submitted to Nature Communications. This manuscript is well suited to the topics of Nature Communications as it uses a combination of relatively new observations to investigate mechanisms governing the response ecosystems to drought. This topic is critical for understanding how the ecosystem will change with droughts expected to increase with climate change. The authors use a combination of remote sensing observations related to ecosystem structure, photosynthesis, evapotranspiration, and water status and seek to separate structural (LAI) influence on ecosystem function during drought from physiological (stomata etc.). I appreciate this comprehensive and multivariate approach to address this topic with observations. However, a more complete treatment of uncertainty is required to know how strong the evidence is for determining physiological drivers of drought response in ecosystems. In addition, further discussion of how what is learned about the physiological response to drought should be considered and applied to Earth System models would help more strongly motivate the paper. Detailed comments below.

Reviews of [figures] and [lines] below.

Major revisions:

[Drought Detection] The method used for drought detection selects for the lowest soil moisture year but does not address the variance of soil moisture or how much of an extreme that minimum year is compared to variability. Why not use a definition more similar to literature that detects drought as an anomalously dry year? There is likely a large overlap in points here, but as it is the detection method would also select for the driest year, even if that year was very similar to all other years. An example of creating a drought index from soil moisture can be found in Sheffield et al. 2004.

[Figure 2] The uncertainty in Figure S6 and S7 should be displayed on Figure 2 to show whether these anomalies are significant. Ideally the uncertainty would be a standard error that accounts for a reduction in degrees of freedom due to spatial autocorrelation. In addition, it would be useful to know how large of an anomaly these are in comparison to the rest of the 4 year dataset.

[223 – 224] $R^2 > 0$ is a very permissive threshold for an empirical model – what is the justification for this threshold and does it impact your results? In addition, it appears that the random forest model generally only explains $< 50\%$ of the variance in the observed remote sensing – this should be remarked on in the main text. What is the characteristic of the remainders?

[61 – 64][226 – 227] What about the patterns due to structural change? Judging from Figure 2, they are also quite similar to the overall pattern of changes in at least SIFrel.

[256 – 258] While the effort to separate the structural LAI effect from the physiological effect is worthwhile and informative, I think referring to the remainder as the physiological effect overstates the precision of the analysis considering caveats such as this regarding the influence of VPD which is beyond physiology. In addition, there may be properties of structure that are not captured by LAI (e.g. leaf angle as noted in [274 – 280]). Further discussion of the mechanisms behind LAI change over 8 day and monthly time periods would help capture the full range of structural impacts that are possible.

[Figure 3][404 – 406] In the conclusion, it is stated that the “...observed drought response of photosynthesis and evapotranspiration are related to, and largely driven by, ... vegetation physiology.” However, I do not see analysis of the relative strength of the physiological pattern vs. the structural pattern. From Figure 3 it appears that the physiological pattern does not account for significant variability in both SIFrel and ET. Adding a row that showed the percentage of the signal that is physio compared to structural (or total) would be informative here. In addition, uncertainty for the anomaly should be displayed somehow, consider adding some hatching for anomalies that are (or are not) significantly different that zero.

[290 – 292] Further comment on what might explain the 65% of the variance that is not explained would be useful. Is this error that is focused in a particular region?

[Figure 4][292- 295] For the SHAP analysis, any cross correlation between predictors will make separating the importance of those predictors (e.g. Aridity vs. Radiation in Figure 4a). Considering this, is it possible to state that one factor is more important than another when their importance is so similar outside of a few leading factors in specific cases?

[301 – 302] This sentence does not match my reading of Figure 4d – wouldn't the “duration of the period during which soil moisture is below seasonal average before drought peak” be (Dev.) Soil Moisture or (Dev.) Duration rather than the leading value of Soil Moisture? I suggest clarifying the labeling or correcting the sentence.

[SCOPE Simulations] Are each of LUE, Gs, and WUE calculated as leaf level values? Wouldn't

running SCOPE without the LAI anomalies be a more complete separation of structural change (held in a non-drought state) and the physiological change in response to drought?

[Figure 5] I suggest some demarcation of where the model does and doesn't agree with (maybe just the sign) of the observed pattern.

[407 – 408] 'Confirms' is too strong for the evidence. Perhaps "shows similar patterns in the hypothesized mechanisms".

Minor revisions:

[123][353] I do not think using a model to infer mechanisms should be referred to as 'validation' in this context.

[Figure 2] It is also my understanding that Figure 2 is primarily for illustrative purposes. The Dry and Wet areas that are aggregated span multiple continents and latitude, how coherent are the time series of drought across these dispirit regions? It may be more illustrative and consistent with the rest of your analysis to show a selection of the aridity bins chosen for Figure 3.

[Figure 5] What are the white numbers?

[414 – 418] There was not much in the way of discussion about how determining the physiological responses would help with Earth system models or what the evidence is that Earth system models do not already include sufficient physiological changes under drought.

References

Sheffield, J. (2004). A simulated soil moisture based drought analysis for the United States. *Journal of Geophysical Research*, 109(D24), D24108. <https://doi.org/10.1029/2004JD005182>

Reviewer #1 (Remarks to the Author):

This paper investigates the vegetation physiological response to drought conditions using observational data. The manuscript is very well written, and the methodology section is detailed. I recommend the manuscript for publication with minor revisions.

A1: We thank the reviewer for such positive feedback.

This study uses a comprehensive set of remote sensing data to infer vegetation structural properties and physiological ones, and disentangle them from each other. Using soil moisture data from ERA5 they identify drought periods across the globe, and then apply their methodology to detect vegetation phenological responses. The methodology uses two Random Forest models applied to the observations and is very well illustrated in Figure S1 in addition to the description. To ensure results are not sensitive to the choice of model, authors also 1) use a Multiple Linear Regression model and 2) apply explainable machine learning technique to the random forest model, and results are consistent with the initial approach.

In addition, the SCOPE model is employed coupled with a radiative transfer model to simulate vegetation response to the identified drought conditions, and the output of the model (and the identified mechanisms) are compared to the results from observations (which validates the results). Authors then discuss their findings (Figs 2-4) in terms of vegetation physiological responses to drought, and provide physical explanation on why these results are meaningful (reference to other relevant studies that have looked similar problems are also provided and similarities and differences are discussed).

Overall, the methodology is solid, and the results are justifiable. Certainly, there are assumptions made in the data pre-processing, the methodology and what you can infer from the observations, but they are all reasonable and very well explained (including any caveats). I do not believe that these assumptions would impact the results in any way.

A2: Thanks for the nice summary of our methods. We are happy to see that our methods are understandable and clear for the reviewer.

Minor comments:

- Lines 608-611: This is a key assumption. Please elaborate why you decided to include more predictors to account for the physiological influence (beyond the fact that the data is available)? What if you lower the number of predictors for the physiological response, how this might impact the results? If you have done any sensitivity test on this, please report.

A3: The reviewer raises an interesting point here. We include 5 hydro-meteorological variables (temperature, precipitation, vapor pressure deficit, soil moisture, and incoming shortwave radiation) to estimate the physiological influence, because these variables are main controls of vegetation photosynthesis and transpiration (e.g., Li et al., 2021; Denissen et al., 2020; Novick et al., 2016).

We now add Supplementary Figure 21 to show physiological anomalies disentangled by SHAP values and multiple linear regression with a reduced number of predictors, 3 main predictors (temperature, incoming shortwave radiation, and soil moisture) instead of 5 as in the original manuscript. We find that the reduction of the number of considered hydro-meteorological variables leads to smaller detected physiological drought anomalies in the case of ET, while the difference is less pronounced

for SIF and VOD. This confirms our initial suspicion about the limitation of the approaches used in this extra analysis as well as the reviewers comment that the number of variables affects the magnitude of the detected physiological effects. This is related to the collinearity between LAI and hydro-meteorological anomalies. However, overall the changes in the detected physiological effects are minor such that the supplementary analyses still largely confirm the patterns detected with our main methodology. We describe this new sensitivity test in Lines 658-660.

While such decomposition methods have a potential to overestimate physiological anomalies, we use our main method which is the random forest prediction difference (using only vegetation structure as a predictor and using both structure and hydro-meteorological variables as predictors). We also discuss that our main method avoids the potential underestimate of vegetation structure in the decomposition methods, although it tends to underestimate vegetation physiology if physiology shares changes with structure (Lines 661-662). Nevertheless, using all relevant methods suggest our robust findings.

We add these points in Lines 658-662 (please find corresponding line numbers in the document with tracked changes) as:

"We test the effect of using a lower number of hydro-meteorological predictors in the variance decomposition method and find a reduced magnitude of resulting vegetation physiological patterns for ET. This suggests that the decomposition method is sensitive to numbers of predictors (Fig. 21). Note that our main method of detecting physiological effects as the difference between two random forest models used throughout the manuscript avoids this potential issue."

New Figure S21. Similar as in Fig. S16 but using only 3 hydro-meteorological variables (temperature, incoming shortwave radiation, and soil moisture) to disentangle physiological variations by (a-c) SHAP values on random forests and (d-f) multiple linear regression.

- Proofread the paper. There are some typos here and there.

A4: Sorry for some typos. We have corrected them now in lines 189, 208, 498, and 601.

- It's encouraging to see the code will be shared for the final publication. I encourage you to share the full end-to-end code from data processing to the final results. Being able to reproduce these results is huge benefit to the community.

A5: Thanks for the suggestion. The codes required for reproducing the results and figures in the main text have been deposited at <https://doi.org/10.5281/zenodo.7971319>, as well as the data to run the codes are available at <https://doi.org/10.5281/zenodo.7971170>

Reviewer #2 (Remarks to the Author):

Summary:

This is a review of the manuscript Widespread and complex drought effects on vegetation physiology inferred from space (NCOMMS-23-08597) submitted to Nature Communications. This manuscript is well suited to the topics of Nature Communications as it uses a combination of relatively new observations to investigate mechanisms governing the response ecosystems to drought. This topic is critical for understanding how the ecosystem will change with droughts expected to increase with climate change. The authors use a combination of remote sensing observations related to ecosystem structure, photosynthesis, evapotranspiration, and water status and seek to separate structural (LAI) influence on ecosystem function during drought from physiological (stomata etc.). I appreciate this comprehensive and multivariate approach to address this topic with observations.

B1: We thank the reviewer for this very positive feedback.

However, a more complete treatment of uncertainty is required to know how strong the evidence is for determining physiological drivers of drought response in ecosystems. In addition, further discussion of how what is learned about the physiological response to drought should be considered and applied to Earth System models would help more strongly motivate the paper. Detailed comments below.

B2: Thanks for commenting on the representativeness of illustrated drought response in ecosystems, and on further discussion about potential applications in ESMs. We provide point-by-point responses below (please find the response to the first point in B3, B4, B5, and B8, and the response to the second point in B18).

Reviews of [figures] and [lines] below.

Major revisions:

[Drought Detection] The method used for drought detection selects for the lowest soil moisture year but does not address the variance of soil moisture or how much of an extreme that minimum year is compared to variability. Why not use a definition more similar to literature that detects drought as an anomalously dry year? There is likely a large overlap in points here, but as it is the detection method would also select for the driest year, even if that year was very similar to all other years. An example of creating a drought index from soil moisture can be found in Sheffield et al. 2004.

B3: We thank the reviewer for the suggestion. To account for the anomalousness of the driest years in each grid cell, we additionally detect droughts from normalized anomalies of the employed 40-year soil moisture data where we also select the driest month but this is only considered as a drought if the normalized anomaly is below (drier than) -1.5 standard deviations. Analyzing the physiological drought responses from the droughts and grid cells detected with this alternative approach (new Figure. S18) reveals very similar physiological patterns of SIFrel, ET, and VOD ratio in our main results.

We add these points in Lines 374-380 in the main text as:

"To further test the robustness of the drought detection, we (i) more strictly select severe drought events by checking if the detected driest soil moisture is lower than a threshold of -1.5 standard deviations below the seasonal mean value of the entire 40-year soil moisture ... Figure S18 shows that with a more strict severe drought evaluation method, the remaining grid cells can largely reproduce physiological patterns of SIFrel, ET, VOD ratio under drought."

We add a new supplementary Figure S18.

New Figure S18. (a, b) Timing of drought peaks, and (c, d, e) vegetation physiological response to drought, but this time we only consider grid cells where the minimum of the 1982-2021 monthly soil moisture is lower than -1.5 standard deviations of soil moisture during the entire period.

[Figure 2] The uncertainty in Figure S6 and S7 should be displayed on Figure 2 to show whether these anomalies are significant.

B4.a: We add inter-quartile ranges in new Figure S6 to show spatial variations of data anomalies which are previously shown in Fig. S6 and S7. We would like to clarify that these spatial variations are driven by heterogeneity in soil and vegetation characteristics and hydro-meteorological conditions, and are not necessarily reflecting uncertainties of vegetation drought responses. This is in line with the reviewer's comment in B16 that we use Figure 2 for illustrative purposes, rather than quantifying vegetation drought response with statistical significance.

We add these points in Lines 216-220 in the main text as:

"We further quantify the spatial variability in the vegetation drought response across grid cells with the envelopes in Fig. S2 and find that this is large, underlining the relevance of vegetation and soil characteristics for the local vegetation drought response. Note that this spatial variability does not necessarily reflect the uncertainties related to the assessment of vegetation drought responses."

We add a new supplementary Figure S6 to replace the old Figures S6 and S7.

New Figure S6: Same as in Fig. 2 but with inter-quartile (25 - 75 %) ranges in the grey shade.

Ideally the uncertainty would be a standard error that accounts for a reduction in degrees of freedom due to spatial autocorrelation.

B4.b: We thank the reviewer for this suggestion which helps to measure the variability of vegetation anomalies to drought after accounting for a reduced spatial autocorrelation. We add the standard error (SE) of vegetation anomalies and soil moisture data streams during the drought period in new Figure 2 with shades. The standard error is calculated based on the standard deviation across the anomalies of every third grid cell in latitude and longitude direction, respectively, which is divided by the square root of the number of considered grid cells. Using such a reduced set of grid cells with a distance of 50 km between each other as opposed to all wet or dry grid cells ensures to minimize the effect of spatial autocorrelation. We also re-computed the standard errors using the neighboring grid cells (but still only every third one) yields very similar results; this indicates that the standard error estimation is not affected by the choice of grid cells.

We add the standard error calculation in the figure caption in Lines 263-267:

"Shades in figures denote the mean standard error which is computed based on the standard deviation across the anomalies of every third grid cell in latitude and longitude direction, respectively. Using such a reduced set of grid cells as opposed to all wet or dry grid cells ensures to minimize the effect of spatial autocorrelation."

We update a new Figure 2.

New Fig.2: Evolution of drought-related anomalies of multiple remote-sensing vegetation variables. Drought-affected grid cells in (a) dry regions (aridity > 1) and (b) wet regions (aridity ≤ 1). Grid cells are only considered if data is available for at least 20 out of the 24 displayed time steps before, during and after drought peaks. Results are averaged across grid cells; Results for dry regions are presented in (c, e, g, i) and for wet regions are presented in (d, f, h, j). (c, d) LAI and NIRv, (e, f) SIF and relative SIF (SIFrel), (g, h) VOD at midday, midnight and the ratio between them (VOD ratio), (i, j) ET and soil moisture. All vegetation variables are shown as anomalies, except for soil moisture in (i, j) which is presented in absolute values to indicate the actual water amount. Shades in figures denote the mean standard error which is computed based on the standard deviation across the anomalies of every third grid cell in latitude and longitude direction, respectively. Using such a reduced set of grid cells as opposed to all wet or dry grid cells ensures to minimize the effect of spatial autocorrelation.

In addition, it would be useful to know how large of an anomaly these are in comparison to the rest of the 4 year dataset.

B4.c: To address the reviewer's question about the magnitude of vegetation anomalies in Fig. 2, we now add a new Figure S7 to present normalised anomalies which are vegetation anomalies divided by the respective standard deviation for each grid cell. The result shows that during the drought period, NIRv features larger variability than LAI, SIF than SIFrel, and midday or midnight VOD than VOD ratio, whereas ET and LAI variability is similar. Soil moisture absolute values are also divided by their standard deviation and results show a larger variability of soil moisture reductions in wet regions than dry regions.

We add these points in Lines 220-223 as:

"The normalised anomalies of vegetation drought trajectories are presented in Fig. S7. The result shows a larger magnitude of NIRv and SIF anomalies compared to other vegetation variables, and soil moisture reductions show larger variability in wet than dry regions."

We add a new supplementary Figure S7.

New Figure S7: Similar to Fig. 2 but presenting the normalised vegetation anomalies using the anomalies divided by respective standard deviation for each grid cell. Soil moisture absolute values are also divided by their standard deviation.

[223 – 224] $R^2 > 0$ is a very permissive threshold for an empirical model – what is the justification for this threshold and does it impact your results? In addition, it appears that the random forest model generally only explains $< 50\%$ of the variance in the observed remote sensing – this should be remarked on in the main text. What is the characteristic of the remainders?

B5: We note that the random forest model performance is typically low when predicting anomalies of global vegetation indices compared to time series that include the seasonal cycles (Kraft et al., 2019; Li et al., 2021). Recent literature has shown that despite the low performance of the anomaly prediction, it can still be efficiently used to study relationships between predictor variables and targets

(Kraft et al., 2019; Li et al., 2021). Regions with Out-of-bag (OOB) $R^2 < 0$ are associated with very low vegetation variability, frequent human management such as tree logging, or poor data quality in predictor or target variables. To address the reviewer's comment, and to better clarify the performance of the random forests, we (i) add the references in the main text as well as the explanation, and (ii) provide different thresholds of OOB R^2 in new Figure S10. The new results using OOB R^2 thresholds of 0.1 and 0.2 show that the pattern of vegetation physiological response to drought is largely unchanged. In the case of considering only grid cells with OOB R^2 greater than 0.2, the weak negative SIFrel physiological changes are slightly shifted to weak positive in very wet regions while very few grid cells remain in this aridity class.

We add these points in the revised manuscript at Lines 249-255 as:

"Since the random forest model performance is rather limited when predicting anomalies of global vegetation indices compared to the prediction of time series that include the seasonal cycles (Kraft et al., 2019; Li et al., 2021), we also present vegetation physiology patterns using different thresholds of out-of-bag R^2 (i.e. 0.1 and 0.2). Results indicate the physiological anomalies are a bit more pronounced but overall largely unchanged, except for SIFrel physiology in very wet regions with an out-of-bag R^2 threshold of 0.2 due to the low number of available grid cells (Fig. S10)."

We add a new supplementary Figure S10.

New Figure S10. (a-c) Same as Fig.3 (d-f) with white numbers denoting numbers of grid cells belonging to each aridity group. (d-i) Similar to Fig.3 (d-f) but keeping regions with random forest out-of-bag $R^2 > 0.1$ (a-f) and out-of-bag $R^2 > 0.2$ (g-i).

[61 – 64][226 – 227] What about the patterns due to structural change? Judging from Figure 2, they are also quite similar to the overall pattern of changes in at least SIFrel.

B6: We note that LAI and SIFrel are different when comparing their trajectories during the drought development periods in wet regions. To better quantify vegetation structural changes under drought, we add a new Figure S9 to present the respective structural components of SIFrel, ET, and VOD ratio. In wet regions, SIFrel structure has positive anomalies whereas SIFrel physiology has weak to negative anomalies. SIFrel and ET have similar structural changes while VOD ratio does not, because VOD ratio by construction is largely insensitive to structural changes at a daily scale (Zhang et al., 2019). All variable structural anomalies have considerably smaller magnitude changes compared to physiological anomalies in Figure 3.

We add these points in Lines 241-247 in the main text as:

"The magnitudes of physiological changes in SIFrel, ET, and VOD ratio are larger than the respective structural changes (Fig. S9). In wet regions, structural and physiological changes of SIFrel have different signs which indicates the decoupling between structure and photosynthetic rate, while for the case of ET, structural and physiological anomaly patterns are similar with negative anomalies in dry regions and positive anomalies in wet regions. Structural anomalies for VOD ratio do not have a clear pattern, and the anomaly magnitude is very small, due to very few structural signals remaining in the ratio."

We add a new supplementary Figure S9.

New Figure S9. Similar to Fig.3 (d-f) but for changes related to vegetation structural changes as estimated from LAI for (a) SIFrel, (b) ET, and (c) VOD ratio.

[256 – 258] While the effort to separate the structural LAI effect from the physiological effect is worthwhile and informative, I think referring to the remainder as the physiological effect overstates the precision of the analysis considering caveats such as this regarding the influence of VPD which is beyond physiology. In addition, there may be properties of structure that are not captured by LAI (e.g. leaf angle as noted in [274 – 280]). Further discussion of the mechanisms behind LAI change over 8 day and monthly time periods would help capture the full range of structural impacts that are possible.

B7: We agree with the reviewer on the potential influence of VPD on adding additional uncertainties in our assessment of vegetation physiological responses as we have noted in Lines 256-258 in the previous version of the manuscript. In fact, our main finding about the strong vegetation physiological downregulation in water-limited regions is consistent among different observation-based physiological variables, SIFrel, ET, and VOD ratio. VOD ratio supports ET results regarding vegetation physiological responses to drought which are both largely associated with changes in stomatal conductance. These, in turn, support our results' robustness despite a direct impact of VPD on ET.

Another uncertainty is related to the leaf angle distribution which is a part of vegetation structure, but the data are not available at the global scale with temporal dynamics. At the site level, leaf angle changes influence the fraction of absorbed photosynthetically active radiation and leaf temperature and complement vegetation structural changes with LAI. At the ecosystem level, to what extent the leaf angle could decouple with LAI is an interesting question but so far not feasible to answer as limited by observations (Yang et al., 2023). Also, as the reviewer mentions, the representation of LAI on vegetation structure could also differ across different temporal scales. To test if using LAI underestimates vegetation structural changes, we have replaced LAI by NIRv which is an independent

vegetation spectral index sensitive to both changes in LAI and leaf angle distribution. We find the main results are largely unchanged, suggesting the capacity of using LAI in representing most synchronized vegetation structural changes (Figure S12).

We clarify these points in the result section, and summarize the relevant limitation in the conclusion as:

"Nevertheless, both VOD ratio and ET indicating stronger downregulation of physiological controls in dry regions shows the robustness of our results, despite a direct impact of meteorology on ET" (Lines 280-282).

"Whereas MODIS LAI includes a clumping correction (Yan et al., 2016), the leaf angle distribution is not considered, and leaf angle distribution data is not available at a global scale ... To further test if using LAI could underestimate vegetation structural changes, we replace LAI by NIRv which is an alternative indicator of vegetation structure, and thus can avoid the simplification of leaf angle distribution in the application of LAI (Fig. S12; Zeng et al., 2019). Overall, this yields similar patterns of physiological controls, together suggesting the capacity of using LAI in representing most synchronized vegetation structural changes" (Lines 297-304).

"Despite that our method simplifies vegetation structural changes and cannot separate potentially direct meteorological-driven signals beyond physiology in the case of ET, our result robustness is addressed by synthesizing multiple observations. The physical-based model SCOPE suggests similar physiological patterns in the diagnosed physiological changes during drought" (Lines 450-453).

[Figure 3][404 – 406] In the conclusion, it is stated that the "...observed drought response of photosynthesis and evapotranspiration are related to, and largely driven by, ... vegetation physiology." However, I do not see analysis of the relative strength of the physiological pattern vs. the structural pattern. From Figure 3 it appears that the physiological pattern does not account for significant variability in both SIFrel and ET. Adding a row that showed the percentage of the signal that is physio compared to structural (or total) would be informative here. In addition, uncertainty for the anomaly should be displayed somehow, consider adding some hatching for anomalies that are (or are not) significantly different that zero.

B8: The reviewer raises an interesting suggestion about quantifying the relative proportions of vegetation physiological changes, and about adding a significance test of observed vegetation physiological anomalies in Figure 3. We update Figure 3 with one additional row in the bottom to show the proportions of physiological anomalies compared to the total vegetation anomalies. We can find that the physiological patterns account for a large part of total anomalies. This confirms our conclusion of "Observed drought response of photosynthesis and evapotranspiration are related to, and largely driven by, unique changes of vegetation physiology." Note that this is also confirmed in the new Figure S9 in our response B6 where we see relatively small magnitude changes of structural components under drought compared to physiological components.

We also report the significance of observed physiological anomalies compared to the physiological changes in the non-drought years during the same seasons (i.e. \pm 16 day time steps).

We add these points in the main text as:

"physiological changes explain 60-97% of the overall functional drought responses in Fig. 3 (d-e)." (Lines 235-236)

"The magnitudes of physiological changes in SIFrel, ET, and VOD ratio are larger than the respective structural changes (Fig. S9)" (Lines 241-242)

"The numbers in the bottom rows denote the median ratio between vegetation physiological anomalies and total (physiological + structural) anomalies across the entire drought period" (Lines 263-265)

"Black dots in each bin denote that in more than 60% of the grid cells, the vegetation physiological anomaly is significantly different (95 % confidence) from a random sample of 1000 samples from the same season (i.e. ± 16 -day time steps) of a non-drought year" (Lines 265-267)

We update a new Figure 3.

New Fig.3: Vegetation functional and physiological responses to drought. Ecosystem functioning as reflected by (a) SIFrel, (b) ET, and (c) VOD ratio anomalies. Ecosystem physiology is estimated as the components of (d) SIFrel, (e) ET, and (f) VOD ratio anomalies remaining after removing the LAI-related variations. Each aridity-drought period box shows the median value across corresponding grid cells and time windows. Aridity classes are chosen to yield a similar number of grid cells in each group on the x-axis. The numbers in the bottom rows denote the median ratio between vegetation physiological anomalies and total (physiological + structural) anomalies across the entire drought period. Black dots in each bin denote that in more than 60% of the grid cells, the vegetation physiological anomaly is significantly different (95 % confidence) from a random sample of 1000 samples from the same season (i.e. ± 16 -day time steps) of a non-drought year.

[290 – 292] Further comment on what might explain the 65% of the variance that is not explained would be useful. Is this error that is focused in a particular region?

B9: We thank the reviewer for this question. Firstly, we predict vegetation physiological anomalies during drought extremes, and the prediction performance is commonly limited even using machine learning approaches, because the seasonality related to climatology and phenology is removed (please also see our response B5). Therefore, we would like to clarify that this is not an error. Secondly, observations of leaf area index, vegetation photosynthesis, evaporation, and vegetation water content have their respective uncertainties and spatial-temporal gaps caused by different characteristics of satellite orbits and retrieval uncertainties of satellite signals. Thirdly, hydrometeorological reanalysis data we employ also contain uncertainties related to model representations, data assimilation and uncertain in-situ measurements. Furthermore, different availability and accessibility of deep water sources such as groundwater, for which no reliable global gridded observations are available in terms of the spatial-temporal scales of our study to our knowledge, can introduce uncertainties here.

We add these points in Lines 674-681:

"The attribution analysis can generally explain over 0.35% of the spatial variability of each physiological variable. The remaining 65% that cannot be explained by the random forest model are potentially related to uncertainties in observations of leaf area index, vegetation photosynthesis, evaporation, and vegetation water content, and also to uncertainties in the hydrometeorological reanalysis data. Furthermore, different availability and accessibility of deep water sources such as groundwater, for which no reliable global gridded observations are available in terms of the spatial-temporal scales of our study to our knowledge, can introduce uncertainties here."

[Figure 4][292- 295] For the SHAP analysis, any cross correlation between predictors will make separating the importance of those predictors (e.g. Aridity vs. Radiation in Figure 4a). Considering this, is it possible to state that one factor is more important than another when their importance is so similar outside of a few leading factors in specific cases?

B10: We thank the reviewer for this comment. Since SHAP importance tends to split variable importance between variables with strong collinearity, we add an additional test using spearman correlation which can assess variable importance individually and independently from each other. We first calculate the correlation coefficient for each predictor and each vegetation physiological variable, and then determine the variable importance by comparing different correlation coefficients (new Figure S14). We compare correlation results and SHAP importance, and mark the top 5 variables in SHAP importance that are consistently top 5 ranking in the correlation results in the revised Figure 4. We note that the main findings of first-order controls of aridity and tree cover fraction as well as main meteorological anomaly controls are robust in different methods in regulating vegetation physiology during drought development periods. In drought recovery periods, instantaneous soil moisture, VPD, and some more meteorological drivers are robust in regulating spatial variability of vegetation physiology.

We add these points in Lines 336-343 in the main text:

"We also apply spearman correlation as an alternative method of assessing and ranking the variable importance. For this purpose we compute the absolute correlation coefficient between each considered explained variable and vegetation physiological variable (Fig. S14). We find that the first-order

controls of vegetation physiology during drought development periods (i.e. aridity, tree cover fraction, and main meteorological anomaly controls) are consistently identified in the correlation analysis, and in drought recovery periods, instantaneous soil moisture, VPD, and a few more meteorological drivers are robust in regulating spatial variability of vegetation physiology."

We update Figure 4 and write " * denotes the variables in the top 5 ranking in SHAP importance results are consistent with correlation results in Figure S14" in the caption.

We add a new supplementary Figure S14.

New Fig.4: Exploring drivers of global patterns of vegetation physiological anomalies under drought. Considered drivers include mean climate and vegetation characteristics (in red), drought-related hydro-meteorological anomalies and drought duration (in blue). Results show their relevance in explaining the spatial variability of anomalies in (a) SIFrel physiology, (b) ET physiology and (c) VOD ratio during drought development. (d-f) Similar as in (a-c) but for drought recovery periods where we consider drought-development (Dev.) and recovery (Recov.) related drought duration and hydro-meteorological anomalies. The unit of relative importance is the same for each physiological variable. Radiation refers to incoming shortwave radiation. * denotes the variables in the top 5 ranking in SHAP importance results are consistent with correlation results in Figure S14.

New Figure S14: Similar to Fig. 4 but using spearman correlation for each predictor and vegetation physiological variable to determine variable importance ranking.

[301 – 302] This sentence does not match my reading of Figure 4d – wouldn't the "duration of the period during which soil moisture is below seasonal average before drought peak" be (Dev.) Soil Moisture or (Dev.) Duration rather than the leading value of Soil Moisture? I suggest clarifying the labeling or correcting the sentence.

B11: Thanks for pointing this sentence out. We agree that this sentence is misleading here. The duration of the drought development is one of the dominant controls of the physiological component of SIFrel. Drought duration is associated not only with the severity of soil moisture depletion but also with other climate conditional changes. We now correct the sentence as "The duration of the drought development is one of the dominant controls of the physiological component of SIFrel" in Line 327.

[SCOPE Simulations] Are each of LUE, Gs, and WUE calculated as leaf level values? Wouldn't running SCOPE without the LAI anomalies be a more complete separation of structural change (held in a non-drought state) and the physiological change in response to drought?

B12: Light use efficiency and water use efficiency are calculated using canopy-integrated photosynthesis, absorbed photosynthetically active radiation, and evapotranspiration, at the site level. Stomatal conductance is the model output also at the site level. These model outputs are simulated firstly as leaf-level values and then are integrated over the canopy layer to calculate top-of-canopy vegetation physiology, as well as water, energy, and carbon fluxes.

We consider that the SCOPE simulations excluding LAI anomalies would be inadequate due to the nature of the model scheme. In the SCOPE model, LAI is not only used as a parameter scaling photosynthesis and transpiration, it also strongly controls radiative transfer and aerodynamics, and hence the energy balance through the vertical profile. Therefore, not representing LAI anomalies could lead to, for example, the unrealistic high levels of photosynthetically active radiation absorbed by canopy (aPAR), forcing the model to deplete fluorescence and to transpire water at exaggerated rates. This would lead to spuriously strong physiological responses that would overestimate our approach capability to separate them from structural changes. To disentangle physiological response in the model and to not introduce additional biases of parameters changes due to the energy balance, we repeated the simulations deactivating the photosynthesis and energy balance modules, so that the simulated (dark-adapted) fluorescence is purely scaled by aPAR but with no physiological down-regulation. The difference between dark-adapted and physiologically-regulated (light-adapted) fluorescence is the simulated vegetation physiological response. The simulated SIFrel physiological responses match those found for the SIFrel physiological extracted from TROPOMI. The simulated SIFrel as well as some more model outputs of vegetation physiological parameters, such as stomatal conductance and light use efficiency, all together suggest the capability of our approach to disentangle vegetation physiological from structural changes.

We add these points in Line 707-711 as:

"SCOPE outputs are simulated firstly as leaf-level values and then are integrated over the canopy layer."

"We use the difference between dark-adapted and light-adapted fluorescence to validate our approach capability to disentangle SIFrel physiology. Also, other vegetation physiological patterns from model outputs such as stomatal conductance and light use efficiency reasonably support our disentangled physiological regulation from vegetation observations under drought."

[Figure 5] I suggest some demarcation of where the model does and doesn't agree with (maybe just the sign) of the observed pattern.

B13: We thank the reviewer for this point. We have discussed the mismatch between observed and SCOPE-simulated physiological patterns during the drought recovery periods in Lines 370-378 in the previous version of the manuscript. We have explained that SCOPE accounts for drought stress through VPD deficits while not through soil moisture deficits, and thus the slowly recovered soil moisture could result in a slow recovery of vegetation physiological changes but is not visible in the model simulations.

In terms of drought development periods, we now add more discussions. The overall contrasting patterns of vegetation physiology between wet and dry regions are consistent in models and observations, while sub-arid and arid regions show slight differences when comparing model results with observations due to potential uncertainties with limited sampling data. For this, we note that the simulations are not to reproduce observations accurately but to produce a comparable variability of vegetation responses to drought.

Overall, despite the potential model uncertainties, we note that the aim of the simulations was not mimicking the observations, which was prevented by model and data limitations, but to validate the capability of our approach to disentangle physiological from structural responses.

We add discussions in Lines 397-402 as:

"Observation-based physiological effects are similar in this subset of grid cells compared to all previously considered areas in terms of the overall contrasting patterns of vegetation physiology between wet and dry regions during the drought development period (Fig. S20; Fig. 5 d-f), while sub-arid and arid regions show slight differences when comparing model results with observations due to potential uncertainties due to model structure and limited sampling data."

[407 – 408] ‘Confirms’ is too strong for the evidence. Perhaps “shows similar patterns in the hypothesized mechanisms”.

B14: We agree. The sentence has now been adapted as "The physical-based model SCOPE suggests similar physiological patterns in the diagnosed physiological changes during drought" in Lines 452-453.

Minor revisions:

[123][353] I do not think using a model to infer mechanisms should be referred to as ‘validation’ in this context.

B15: We agree. These sentences have now been adapted as "Finally, we use the Soil Canopy Observation of Photochemistry and Energy flux (SCOPE) model to simulate the vegetation drought response and underlying physiological changes, and hence enable a mechanistic interpretation of our disentangled vegetation physiology" in Line 124, and "This allows us to mechanistically understand the diagnosed physiological signals from observations" in Line 391.

[Figure 2] It is also my understanding that Figure 2 is primarily for illustrative purposes. The Dry and Wet areas that are aggregated span multiple continents and latitude, how coherent are the time series of drought across these dispirit regions? It may be more illustrative and consistent with the rest of your analysis to show a selection of the aridity bins chosen for Figure 3.

B16: We thank the reviewer for this comment. Please note that the spatial variability of vegetation trajectories during the course of drought in Figure 2 is large as shown in new Figure S6. For a better visualisation in Figure 2, we have separated the results into two kinds, the wet and the dry regions, with relatively similar numbers of grid cells. We believe that the figure is already efficient for readers to understand our findings of vegetation physiological responses to drought, so we prefer to not add additional figures which provide little new information.

[Figure 5] What are the white numbers?

B17: We thank the reviewer for asking this point. The white numbers denote numbers of grid cells belonging to a certain aridity group. We add it now in the figure caption as "**The white numbers denote numbers of grid cells belonging to a certain aridity group**" in Line 438.

[414 – 418] There was not much in the way of discussion about how determining the physiological responses would help with Earth system models or what the evidence is that Earth system models do not already include sufficient physiological changes under drought.

B18: We thank the reviewer for this point. Our work about the vegetation physiological response to drought helps to better understand vegetation-climate coupling processes and mechanisms and indicates the capability of current up-to-date satellite data in capturing vegetation physiological responses. Since Earth system models simulate vegetation physiology with large uncertainties under drought (Stocker et al., 2019), these results open new opportunities to improve simulations of vegetation dynamics in Earth system models, through better parameterisations of vegetation physiology such as photosynthetic capacity (Chen et al., 2022), or through upgrading the respective model structure such as incorporating sufficient soil water stress on vegetation physiology (Trugman et al., 2018).

We add these points in Lines 457-463 in the conclusion as:

"This is essential given the present uncertainties in simulated large-scale ecosystem drought responses (Stocker et al., 2019). In this context, disentangling physiological and biophysical vegetation responses enables a better characterization of these distinct vegetation response pathways to consequently reflect their interplay more accurately in Earth system models through better parameterisations of vegetation physiology such as the photosynthetic capacity (Chen et al., 2022), or through upgrading the respective model structure such as incorporating sufficient soil water stress on vegetation physiology (Trugman et al., 2018)."

References

Sheffield, J. (2004). A simulated soil moisture based drought analysis for the United States. *Journal of Geophysical Research*, 109(D24), D24108. <https://doi.org/10.1029/2004JD005182>

References

Kraft, B., Jung, M., Körner, M., Koirala, S. & Reichstein, M. (2022). Towards hybrid modeling of the global hydrological cycle. *Hydrol. Earth Syst. Sci.* 26, 1579–1614.

Li, W., Migliavacca, M., Forkel, M., Walther, S., Reichstein, M., & Orth, R. (2021). Revisiting global vegetation controls using multi - layer soil moisture. *Geophysical Research Letters*, 48(11), e2021GL092856.

Denissen, J. M., Teuling, A. J., Reichstein, M., & Orth, R. (2020). Critical soil moisture derived from satellite observations over Europe. *Journal of Geophysical Research: Atmospheres*, 125(6), e2019JD031672.

Novick, K. A., Ficklin, D. L., Stoy, P. C., Williams, C. A., Bohrer, G., Oishi, A. C., ... & Phillips, R. P. (2016). The increasing importance of atmospheric demand for ecosystem water and carbon fluxes. *Nature climate change*, 6(11), 1023-1027.

Zhang, Y., Zhou, S., Gentine, P., & Xiao, X. (2019). Can vegetation optical depth reflect changes in leaf water potential during soil moisture dry-down events? *Remote Sensing of Environment*, 234, 111451.

Yang, X., Li, R., Jablonski, A., Stovall, A., Kim, J., Yi, K., ... & Lerdau, M. (2023). Leaf angle as a leaf and canopy trait: Rejuvenating its role in ecology with new technology. *Ecology Letters*.

Stocker, B.D., Zscheischler, J., Keenan, T.F. et al. (2019). Drought impacts on terrestrial primary production underestimated by satellite monitoring. *Nat. Geosci.* 12, 264–270.
<https://doi.org/10.1038/s41561-019-0318-6>

Chen, J. M., Wang, R., Liu, Y., He, L., Croft, H., Luo, X., Wang, H., Smith, N. G., Keenan, T. F., Prentice, I. C., Zhang, Y., Ju, W., and Dong, N. (2022). Global datasets of leaf photosynthetic capacity for ecological and earth system research, *Earth Syst. Sci. Data*, 14, 4077–4093,
<https://doi.org/10.5194/essd-14-4077-2022>.

Trugman, A. T., Medvigy, D., Mankin, J. S., & Anderegg, W. R. L. (2018). Soil moisture stress as a major driver of carbon cycle uncertainty. *Geophysical Research Letters*, 45(13), 6495-6503.

REVIEWERS' COMMENTS

Reviewer #2 (Remarks to the Author):

We complement the authors is a thorough response to my comments, I recommend the paper for publication and look forward to it being in press.

Note in line 691, it is stated that "the attribution analysis can generally explain over 0.35%..." I believe that this is a typo and should read as "35%".

REVIEWERS' COMMENTS

Reviewer #2 (Remarks to the Author):

We complement the authors is a thorough response to my comments, I recommend the paper for publication and look forward to it being in press.

Note in line 691, it is stated that "the attribution analysis can generally explain over 0.35%..." I believe that this is a typo and should read as "35%".

A1: We thank the reviewer for pointing out the typo. We correct it now in Line 674 in the manuscript with changes tracked.